biomedical engineering/mechanical engineering

venous ulcers, stasis ulcers, finite element, inflammation, glycosaminoglycans, sodium

**Author for correspondence:**
Tamara Reid Bush
e-mail: reidtama@egr.msu.edu

# Influences of sodium and glycosaminoglycans on skin oedema and the potential for ulceration: a finite-element approach

Wu Pan[1], Sara Roccabianca[1], Marc D. Basson[2]
and Tamara Reid Bush[1]

[1]Department of Mechanical Engineering, Michigan State University, 428 South Shaw Lane, Room 2555, East Lansing, MI 48824, USA
[2]Department of Surgery at the University of North Dakota School of Medicine and Health Sciences, Grand Forks, ND, USA

(iD) TRB, 0000-0002-2757-0463

Venous ulcers are chronic transcutaneous wounds common in the lower legs. They are resistant to healing and have a 78% chance of recurrence within 2 years. It is commonly accepted that venous ulcers are caused by the insufficiency of the calf muscle pump, leading to blood pooling in the lower legs, resulting in inflammation, skin oedema, tissue necrosis and eventually skin ulceration. However, the detailed physiological events by which inflammation contributes to wound formation are poorly understood. We therefore sought to develop a model that simulated the inflammation, using it to determine the internal stresses and pressure on the skin that contribute to venous ulcer formation. A three-layer finite-element skin model (epidermis, dermis and hypodermis) was developed to explore the roles in wound formation of two inflammation identifiers: glycosaminoglycans (GAG) and sodium. A series of parametric studies showed that increased GAG and sodium content led to oedema and increased tissue stresses of 1.5 MPa, which was within the reported range of skin tissue ultimate tensile stress (0.1–40 MPa). These results suggested that both the oedema and increased fluid pressure could reach a threshold for tissue damage and eventual ulcer formation. The models presented here provide insights to the pathological events associated with venous insufficiency, including inflammation, oedema and skin ulceration.

# 1. Introduction

Venous ulcers, also known as stasis ulcers, are the most common type of leg ulcers affecting millions of people globally [1–3]. In the USA alone, 25 billion dollars are spent every year for venous ulcer treatment [4]. Venous ulcers are difficult to heal and likely to recur. Indeed, 78% of venous ulcers may reoccur within 2 years [5]. Conventional treatments for venous ulcers are only prescribed after the ulcers have formed. Because the pathophysiology of ulcer formation is not well understood, there are no effective approaches or tests to predict the onset of a venous ulcer [6,7]. To prevent venous ulcer formation and improve treatment, it is important to fully understand the role of inflammation and factors that contribute to inflammation during ulcer formation.

A generally agreed upon mechanism of ulcer formation does not exist [1,8,9]. Venous ulcers are believed to be related to calf muscle pump failure, faulty valves and chronic venous insufficiency [5,8]. These conditions reduce venous function; blood in the lower leg cannot be returned efficiently to the heart, causing it to pool in the lower leg. Such pooling of blood ('stasis') causes inflammation as increased venous back pressure and induces protein leakage across capillaries due to the pressure gradient difference [10]. Such inflammation not only impairs the endothelial barrier and makes it leakier but also leads to oedema of the skin [11–13]. This process of inflammation (followed by oedema) impairs perfusion, causing tissue necrosis and eventually ulceration [12]. To better understand the detailed pathology of venous ulcers, it is important to understand how skin tissue reacts to blood pooling and the inflammatory process. Exploring these biomechanically driven changes through modelling can illuminate potential mechanisms for ulcer formation. The overall goal of this study is to develop a model that simulates the inflammation, determines the internal stresses and pressure of the skin tissue and provides an improved understanding of these mechanisms and their association with venous ulcer formation.

## 1.1. Background

Recent models of the skin focus primarily on the *wound healing* processes, specifically for epidermal healing, dermal extracellular matrix repair, wound contraction and angiogenesis [14–16]. Specifically, in models of wound angiogenesis (formation of new blood vessels), studies report that glycosaminoglycans (GAG) play a significant role in inflammation [17,18]. In several of these models, inflammation and tissue ischaemia are simulated with GAGs and growth factors [15,19–21], and the results indicate that chronic inflammation causes tissue ischaemia [21]. Although several wound healing models exist, there is still a lack of information on *wound formation*.

Venous ulcers are a result of a series of events, starting with blood pooling in the lower legs [22,23], causing blood to leak through the vessel wall [24,25], leading to inflammation. During this inflammation, two biochemical agents, GAGs and sodium, are released as biological responses to the inflammatory process [26–28]. GAGs carry negative charges and attract water into the tissue. This induces an osmotic swelling pressure and leads to tissue swelling, which affects the mechanical properties of the tissues [29–31]. The increased swelling is a result of increased negative charges carried by the GAGs [32–36]. Some research on GAGs and soft tissue damage has been conducted in cardiovascular tissue. Specifically, these studies show that the accumulation of GAGs increases tissue stresses, potentially leading to the tearing or rupture of the aortic wall [36,37]. However, no work has been published on swelling and its relation to venous ulcer formation.

In soft tissue inflammation, an increase in sodium, concurrent with GAG accumulation, is reported [27,38,39]. Titze *et al.* [38] report that sodium increases from 140 to 180–190 mmol l$^{-1}$ with an increase in GAG content, and Crescenzi *et al.* [39] report a 3% increase in sodium content associated with lipoedema. During inflammation, the osmolarity (i.e. sodium concentration) in the extracellular space decreases as the sodium is drawn into the inflamed tissue. When discussing osmolarity changes, it is important to note that our model discussions refer to the changes in osmolarity in the surrounding bath (i.e. the fluid environment surrounding the skin tissue and GAGs) rather than the osmolarity of the interstitial space within the tissue itself. Because of the negative charges, GAGs attract positively charged sodium molecules from the outside bath into the tissue, resulting in water from the external bath drawn across the membrane into the surrounding interstitial space within the tissue, leading to the increased tissue swelling [40–43]. Figure 1 illustrates the analogue process of GAG, sodium and fluid transport through the tissue during inflammation and swelling.

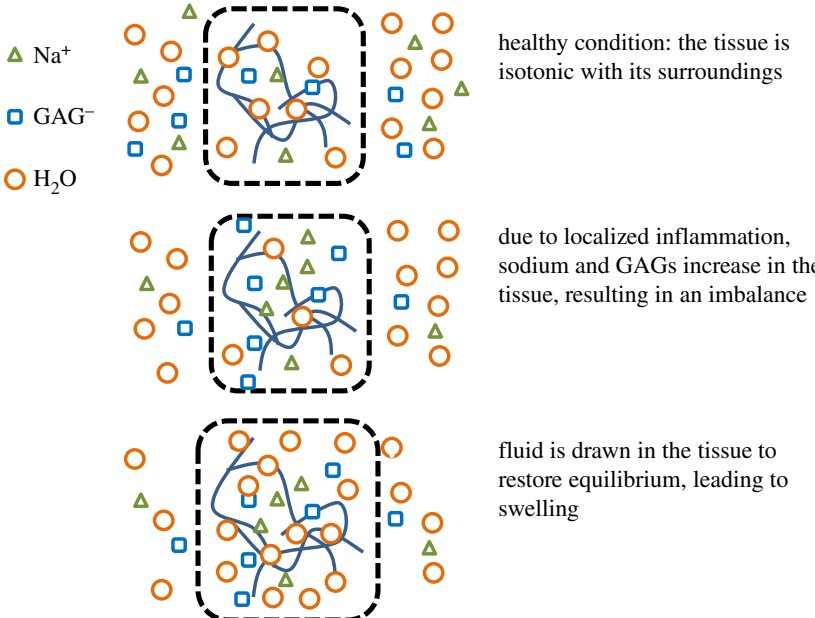

**Figure 1.** Analogue process of GAG, sodium and fluid transportation during inflammation and swelling.

Azeloglu *et al.* [34] investigated the influences of external bath osmolarity in aorta tissue swelling and found that with decreased external bath osmolarity, the tissue exhibited higher swelling and more residual stresses. Lai *et al.* [44] looked at cartilage swelling in different sodium solutions and reported that the lower the sodium content in the bath solution, the higher the swelling strain the tissue experienced.

Therefore, by combining the influences of both GAGs and sodium, our hypothesized physiological pathway from blood pooling to ulcer formation is proposed in figure 2.

In order to validate this pathway and characterize the detailed mechanical changes in the skin tissue, a computational model for the skin was developed. Figure 3 provides a schematic of the broader perspective of modelling blood pooling and skin oedema through changes in GAG and sodium content. As the vein is located in the hypodermis layer of the skin, we expected that the GAG and sodium accumulation would first occur in this layer.

Thus, the specific goals of this study were to (i) compare the effects of different negative charge levels associated with GAGs on skin fluid pressure and elastic stress and (ii) compare the effect of different osmolarity levels (i.e. sodium content) on skin fluid pressure and elastic stress.

## 2. Methods

### 2.1. Model establishment

A three-layer skin model (epidermis, dermis and hypodermis) was developed. The thicknesses of the epidermis, dermis and hypodermis were chosen as 1 mm, 2 mm and 2 mm, respectively; these values were based on the reported literature data from Hendriks *et al.* [45], Groves [46], Flynn *et al.* [47], Zöllner *et al.* [48] and Socci *et al.* [49]. The model in our study was a half-symmetric skin portion with a quarter-spherical-shaped inclusion of GAGs located in the bottom–centre of the hypodermis layer (figure 4). The GAG inclusion identified the region of the skin tissue with accumulated GAG molecules. This terminology was also used by Roccabianca *et al.* [37]. The geometric model was then meshed using hexahedral elements HEX8 (28896 elements per model) and preprocessed in Hypermesh (Hyperworks 12.0, Altair, USA) and later imported into FEBio (v. 2.4.0, University of Utah) for computing and post-processing. The authors also performed a sensitivity analysis. The element number was increased by approximately 30% to generate a more refined mesh, and the results (displacement and stress) did not differ.

Boundary conditions were applied to represent the fact that the model was symmetric with respect to the *YZ* plane, see figure 4*a,b*, and was isolated from a larger skin plane. Namely, we constrained the normal displacement on the plane of symmetry, and normal and shear displacements on the planes in

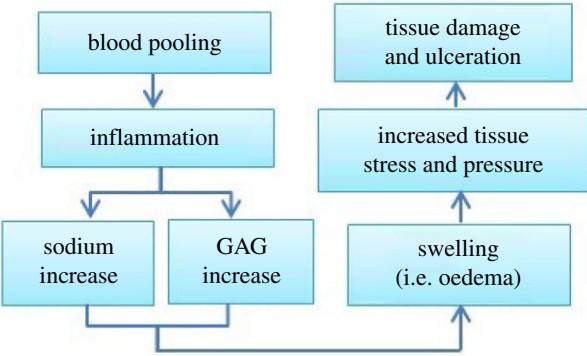

**Figure 2.** Hypothesized physiological venous ulcer formation pathway caused by changes in GAG and sodium content within the GAG inclusion.

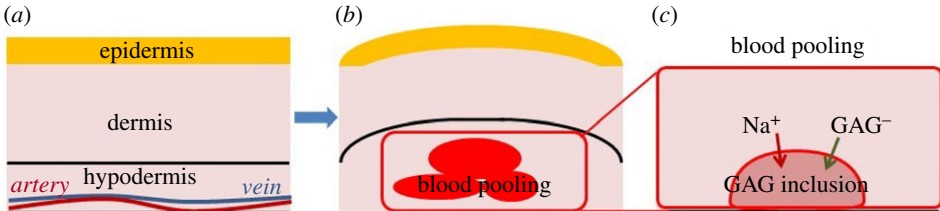

**Figure 3.** The comparisons between (*a*) normal skin and (*b*) oedema in skin due to blood pooling. In venous ulcer patients, the blood pooling occurs because of the inability to return the blood to the heart and faulty venous valves. This pooling leads to inflammation. (*c*) The biochemical response to inflammation is an increase in GAGs and sodium.

contact with surrounding skin. Because we were interested in upward/outward swelling, the top surface, representing the side of the epidermis in contact with air, was allowed free swelling; while the bottom surface, representing the side of the hypodermis in contact with surrounding tissue, was constrained in all directions.

## 2.2. Theoretical pressure and stress analysis

Both fluid pressure and elastic stress affect the total stresses within the tissue [50].

The total stress in our model is a sum of the fluid pressure and elastic stress [34]

$$\sigma = -pI + \sigma^{e}, \tag{2.1}$$

where $\sigma$ is the Cauchy stress and $p$ is the fluid pressure. The negative sign associated with pressure in equation (2.1) is required because the fluid pressure is treated as compression in the stress analysis [51]. $I$ is the identity matrix and $\sigma^{e}$ is the elastic stress.

Therefore, the governing equation to be solved in this model based on conservation of momentum and conservation of mass is

$$\text{div}\,\boldsymbol{\sigma} = -\nabla p + \text{div}\,\boldsymbol{\sigma}^{e} \tag{2.2}$$

and

$$\text{div}\left(\frac{\partial \boldsymbol{u}^{s}}{\partial t} + Q\right) = 0, \tag{2.3}$$

where $\boldsymbol{u}^{s}$ is the porous solid matrix displacement and $Q$ is the fluid flux relative to the solid matrix.

The fluid pressure has been shown to alter tissue compliance and permeability [18,52], while the elastic stress contributes to tissue tearing and damage [36,44]. Therefore, it is important to identify the roles of both fluid pressure and elastic stress in the overall wound formation process. As reported in the literature, the fluid pressure caused by the swelling of accumulated GAGs is defined as [44,51]

$$p = RT\sqrt{(C^{F})^{2} + (\bar{C}^{*})^{2}} - \bar{C}^{*}, \tag{2.4}$$

where $R$ and $T$ are the universal gas constant and absolute temperature, $\bar{C}^{*}$ is the external bath osmolarity and $C^{F}$ is the fixed charge density (FCD) in the current configuration (defined as the final state after

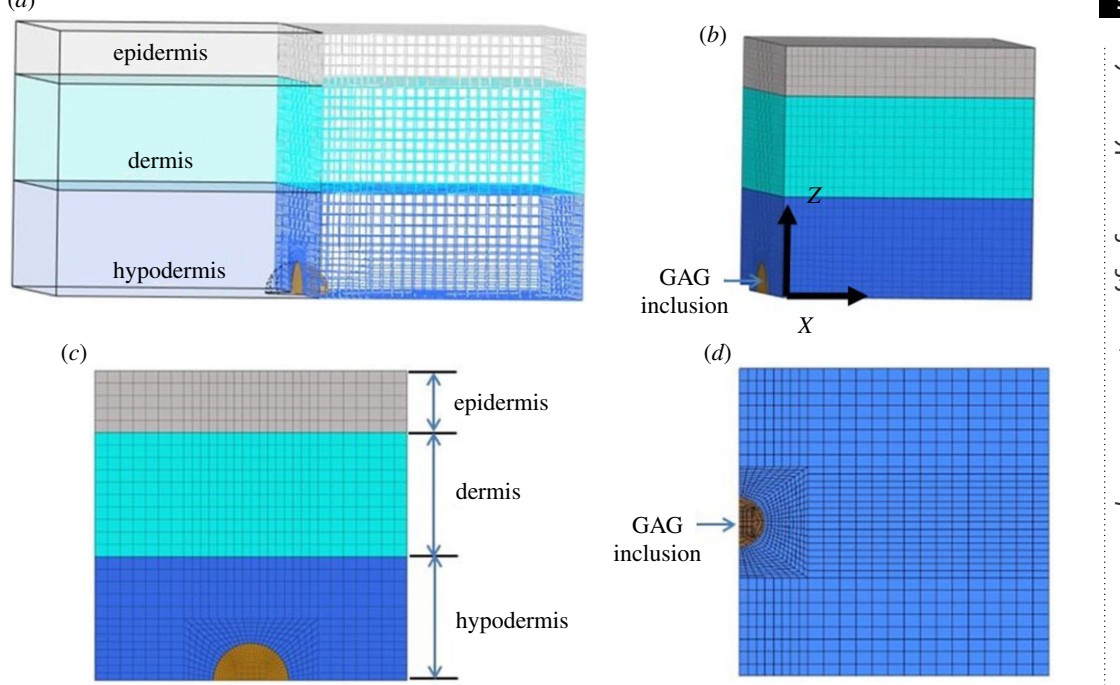

**Figure 4.** The skin finite-element model with the GAG inclusion: (a) the skin is half symmetric to allow the exposure of the GAG inclusion, while the GAG inclusion is quarter-spherical shaped and centrally placed in the hypodermis, (b) oblique view of model with GAG inclusion in the central bottom, (c) left view of the model and (d) bottom view of the model.

swelling). The FCD is defined as the concentration of negative charges fixed in the wet tissue [52]. $C_0^F$ is the FCD at reference configuration (defined as the original state before swelling) and is calculated as

$$C_0^F = \frac{z \cdot c}{M}, \tag{2.5}$$

where $z$ is the number of negative charges carried by the GAGs, $M$ is the molecular weight for the GAGs and $c$ is the milligrams of GAGs per millilitre of water in the skin.

$C^F$ is then calculated as follows [34,51]:

$$C^F = \frac{\varphi_0^w C_0^F}{J - 1 + \varphi_0^w}, \tag{2.6}$$

where $\varphi_0^w$ is the fluid volume fraction and $J = \det F$ is the relative volume (derivative of the deformation tensor $F$) [34,51].

The boundary condition for this model was considered as

$$\boldsymbol{u}^s(0) = 0; \tag{2.7}$$

$$\boldsymbol{u}^s(t) = 0 \quad \text{for constrained surfaces;} \tag{2.8}$$

$$p(0) = RT\sqrt{(C_0^F)^2 + (\bar{C}_0^*)^2} - \bar{C}_0^*; \tag{2.9}$$

and

$$Q(t) = 0. \tag{2.10}$$

The elastic stress endured by the tissue is then calculated as

$$\begin{bmatrix} \sigma_{xx}^e & & \\ & \sigma_{yy}^e & \\ & & \sigma_{zz}^e \end{bmatrix} = \begin{bmatrix} p & & \\ & p & \\ & & p \end{bmatrix} + \begin{bmatrix} \sigma_{xx} & & \\ & \sigma_{yy} & \\ & & \sigma_{zz} \end{bmatrix}. \tag{2.11}$$

In order to quantify the stress localization in the tissue surrounding the GAG inclusion and provide more detailed information about the tissue tensile stress and tissue fluid pressure change under the FCD influences, regions of interest (ROIs) that represented the regions that exhibited the minimum compressive stress (maximum in magnitude) and maximum tensile stress when the surrounding tissue was subjected to the swelling pressure from the accumulated GAGs were identified.

From equation (2.11), the elastic stress can be computed from the fluid pressure and the total stress state and further analysed to evaluate the maximum tensile stress in the ROIs within the model. Given that the shear stress component was zero in equation (2.11), the maximum tensile/compressive stress could also be referred to as the first and third principal stresses, respectively. The authors chose to use the term tensile stress throughout the paper as it was a more descriptive term. FEBio formulated all solid elements in a global coordinate system, and the osmotic pressure was considered isotropic [53]; therefore, in the elastic stress calculation, $\sigma_{xx}^e$, $\sigma_{yy}^e$ and $\sigma_{zz}^e$ were all with respect to the global coordinate.

## 2.3. Model material parameters

Since the inflammation and the resulting skin oedema consisted of both biomechanical and biochemical processes, a mixed model was used, this model coupled materials that represented both mechanical and chemical behaviours. Details are provided in the following sections.

### 2.3.1. Solid material

Skin as a soft biological tissue exhibited large deformations when under small loads [54]; therefore, in accordance with other skin models in the literature, our model also used a nonlinear, hyperelastic, neo-Hookean material to simulate the skin tissue [49,55]. The epidermis and dermis were simulated with the same material properties in the model presented here; the same approach has been reported in several other studies [48,55–57]. The material parameters reported by Buganza Tepole et al. [57] and Zöllner et al. [48] were used for both the epidermis and dermis layers with Young's modulus at 110 kPa and Poisson's ratio at 0.47. For the hypodermis layer, Young's modulus and Poisson's ratio were chosen to be 70 kPa and 0.48, respectively. This fell in the range reported by Li et al. [58] and Gennisson et al. [59].

### 2.3.2. Donnan osmotic swelling material

The charges carried by sodium and GAGs lead to an imbalance of the osmolarity between the external bath and internal regions and induce a Donnan potential [60]. The authors point out that although the Donnan equilibrium is a derivation from the Starling equation [60], the mathematical foundation of this simulation relied on the Donnan effect caused by electrostatic force, rather than the pressure force as illustrated by the Starling equation [13]. Several studies looking at other body tissues have used a swelling material to describe the Donnan effect caused by GAG accumulation within the skin [18,61,62]. In our model, the Donnan swelling material was assigned to the pooled GAG inclusion in the hypodermis as well as the distributed GAGs throughout the skin tissue. To define the Donnan's material model in FEBio, three parameters were necessary: (i) fluid volume fraction; (ii) external bath osmolarity, the osmolarity in the external fluid environment in which the tissue was saturated (figure 5); and (iii) FCD, which is the negative charges carried by the GAGs [44,51]. For the fluid volume fraction, a range between 0.70 and 0.85 was reported in the literature [34,40,63]. In accordance with Bhave & Neilson's [40] work, this value was set to 0.72 for our model. Similar to studies by Azeloglu et al. and Lai et al., the osmolarity values in our study refer to the external bath osmolarity. The physiological osmolarity for normal skin has been reported to be between 270 and 300 mOsm $l^{-1}$ [34,35,64,65]; 280 mOsm $l^{-1}$ was selected as the 'normal' osmolarity level based on these studies, specifically, the work reported by Negoro et al. [65].

For $z$ and $M$ from equation (2.5), the literature reported that GAGs carry two negative charges (2 equivalents, 'Eq') per unit and each GAG unit has a molecular weight of 513 g [52,61]. For $c$, Wiig et al. [18,62] reported the GAG content in the normal skin between 3.7 and 4.2 mg $g^{-1}$ of wet tissue weight and the total tissue water content was 0.6 ml $g^{-1}$ of wet tissue weight; therefore, the average GAG content in total tissue water content was calculated as

$$c = \frac{3.95 \text{ mg g}^{-1}}{0.6 \text{ ml g}^{-1}} = 6.58 \text{ mg ml}^{-1}. \tag{2.12}$$

Hence, based on the above information, the distributed GAG FCD in skin at normal status was derived to be

$$C_0^F = \frac{2 \text{ Eq} \times 6.58 \text{ mg ml}^{-1}}{513 \text{ g}} = 25.7 \text{ mEq l}^{-1}. \tag{2.13}$$

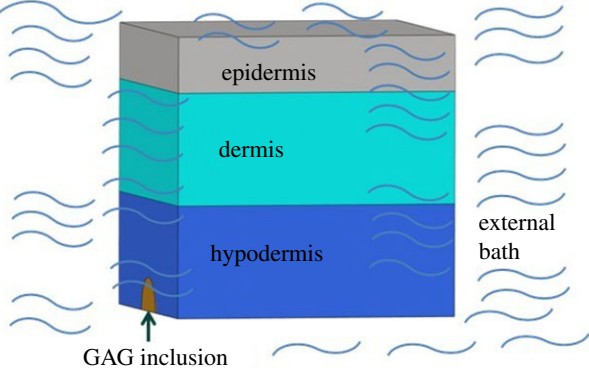

**Figure 5.** External bath of the skin block model. The external bath osmolarity refers to the sodium content within the fluid environment surrounding the skin tissue and GAG inclusion.

## 2.4. Parametric study

A series of parametric studies was conducted to compare the stress and pressure differences in the tissue for different osmolality values of the external bath and different negative charges (i.e. FCD) carried by the accumulated GAGs. Initial simulations evaluated different geometric shapes for the GAG inclusion and different GAG distributions in the skin. Simulation results indicated that the spherical shape for the GAG inclusion along with the uniform GAG distribution produced the largest magnitude of swelling. Therefore, in order to study the 'worst case scenario', a combination of spherical GAG inclusion and uniform GAG distribution in the skin was used.

A total of 10 simulations were performed in FEBio to investigate the influences of five negative charge levels (FCD) associated with the GAG inclusion and five external bath osmolarity levels associated with the GAG inclusion. Each simulation is discussed in the following sections.

### 2.4.1. Different external bath osmolarity levels

For the external bath osmolarity, Azeloglu *et al.* conducted a parametric study on osmolarity ranging from 20 to 2000 mOsm $l^{-1}$, and evaluated the tissue residual stresses in the aorta ring [34]. In our model, a series of external bath osmolarity values were chosen within this range: (i) 50 mOsm $l^{-1}$, (ii) 100 mOsm $l^{-1}$, (iii) 280 mOsm $l^{-1}$, (iv) 500 mOsm $l^{-1}$, and (v) 1000 mOsm $l^{-1}$.

### 2.4.2. Different fixed charge density levels associated with the glycosaminoglycan inclusion

Since the ranges of FCD related to inflammation have not been reported in the literature, the normal FCD level and four values higher than normal were selected: (i) FCD = 25.7 mEq $l^{-1}$, a normal level of FCD for GAGs in the skin tissue as in equation (2.3); (ii) FCD = 150 mEq $l^{-1}$, near the upper limit reported by Roccabianca *et al.* [37]; (iii) FCD = 257 mEq $l^{-1}$, 10 times higher than the normal level; (iv) FCD = 514 mEq $l^{-1}$, 20 times higher than the normal level; and (v) FCD = 771 mEq $l^{-1}$, 30 times higher than the normal level. These values were chosen to parametrically investigate the effects of the GAG inclusion during inflammation and determine the swelling and associated tissue stresses.

# 3. Results

## 3.1. Glycosaminoglycan inclusion fixed charge density comparison

Five simulations were performed for the different FCD. The external bath osmolarity was maintained at 280 mOsm $l^{-1}$ and the FCD in the skin was 25.7 mEq $l^{-1}$. In this way, the only variable was the FCD associated with the GAG inclusion. This was simulated at 25.7, 150, 257, 514 and 771 mEq $l^{-1}$. The total displacement and Von Mises stress of this model at all five FCD levels are presented in figure 6.

The total displacement and the Von Mises stress both increased in the GAG inclusion and the surrounding tissue when the FCD level associated with the GAG inclusion was increased. The increase in FCD caused an increase in the attraction of sodium, which then caused an increase in

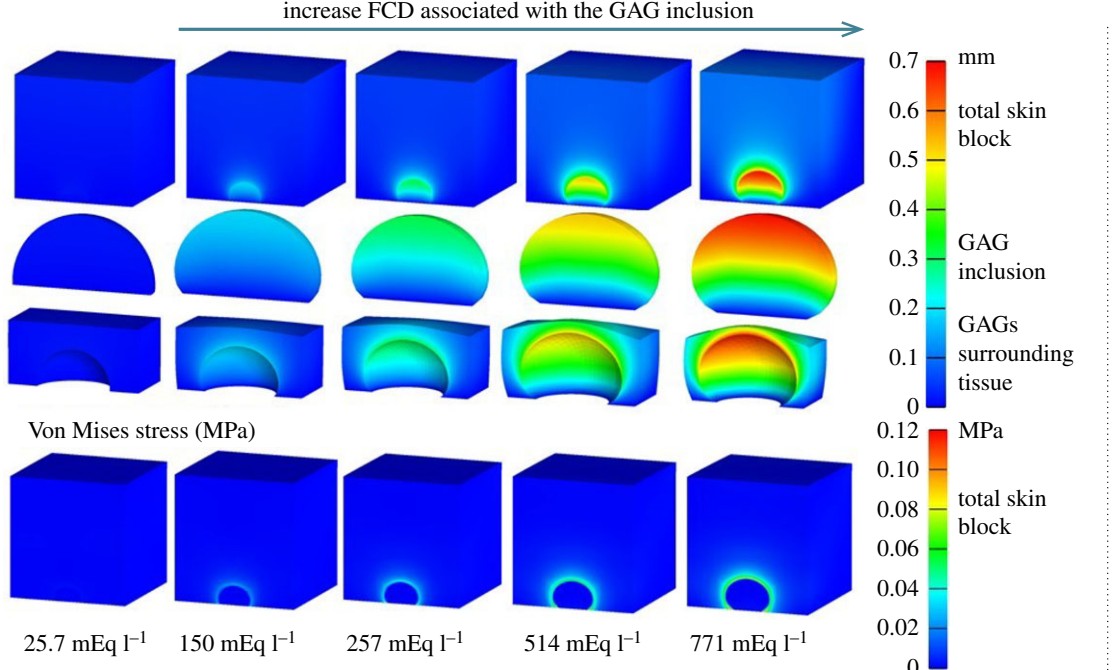

**Figure 6.** Comparison of the influences of the FCD levels associated with the GAG inclusion on the total displacement (mm) and Von Mises stress (MPa). From left to right, five simulations were conducted when the FCD for the inclusion was set to 25.7, 150, 257, 514 and 771 mEq l$^{-1}$. The largest deformation and highest stress concentration was at 771 mEq l$^{-1}$. The displacement and stress increased with the increased FCD of the GAG inclusion.

the attraction of water and increased swelling. As expected, the highest level of swelling occurred when the FCD was 771 mEq l$^{-1}$, which was 30 times higher than the normal FCD level. At the same time, the surrounding tissue of the GAG inclusion was compressed due to the osmotic swelling pressure from the accumulated GAGs. High stresses occurred at the interface between the GAG inclusion and the surrounding tissue at all FCD levels.

From the observation of the stress concentration in figure 6, four ROIs: (i) left, (ii) right, (iii) top, and (iv) centre were chosen for quantitative stress analysis based on the global coordinate system (figure 7). Four elements from each ROI were selected and the averaged stress and fluid pressure were calculated.

The maximum elastic stresses $\sigma_{xx}^e$, $\sigma_{yy}^e$ and $\sigma_{zz}^e$ for the four ROIs were calculated according to equation (2.11) and are reported in figure 8.

When the FCD was 771 mEq l$^{-1}$, the maximum tensile stress was located at the top region of the surrounding tissue, where the tissue interacted with the GAG inclusion and had a magnitude of 1.5 MPa for both $\sigma_{xx}^e$ and $\sigma_{yy}^e$. The maximum stress exhibited a nonlinear increase with the increase in the GAG inclusion FCD. The highest stress was 400 times more when the FCD was 30 times higher than the normal value. When the GAG inclusion FCD was at 150 mEq l$^{-1}$ (the same value reported being used by Roccabianca *et al.* [37]), the maximum elastic stress was 30 times higher than that at normal FCD value.

The fluid pressure within the skin tissue consisted of the osmotic pressure within the GAG inclusion and the interstitial fluid pressure in the surrounding tissue outside of the GAG inclusion. These two pressures were calculated according to equation (2.4) to investigate the change of the skin tissue fluid pressure before and after swelling. Results of the pressure change during swelling are plotted in figure 9.

It can be seen that the osmotic pressure *within the GAG inclusion* decreased after swelling (i.e. the pressure change shows negative values) at all FCD levels. So, before swelling, the osmotic pressure was high, which drew water into the tissue and caused swelling. After the tissue was already swollen, the osmotic pressure reduced to a lower value. This was seen as a 'drop' of osmotic pressure during swelling in figure 9 [28,60]. The definition of osmotic pressure described the minimum pressure a membrane needed to prevent an influx of water [43]. With the increased FCD values, the osmotic pressure within the GAG inclusion decreased even further.

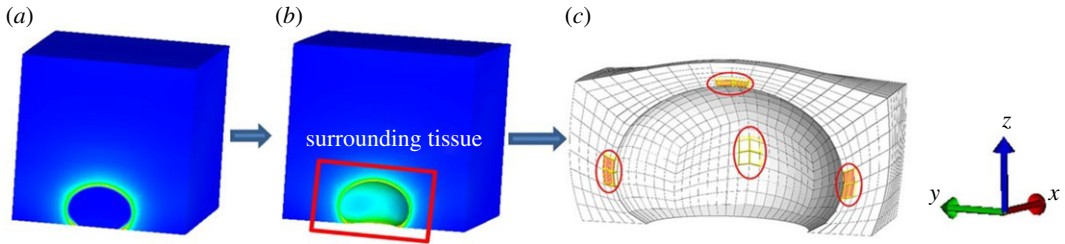

**Figure 7.** The selection of four ROIs in the global coordinate system. (*a*) The whole skin block model, (*b*) the skin block model without the GAG inclusion and (*c*) the surrounding tissue with four ROIs and the coordinate system.

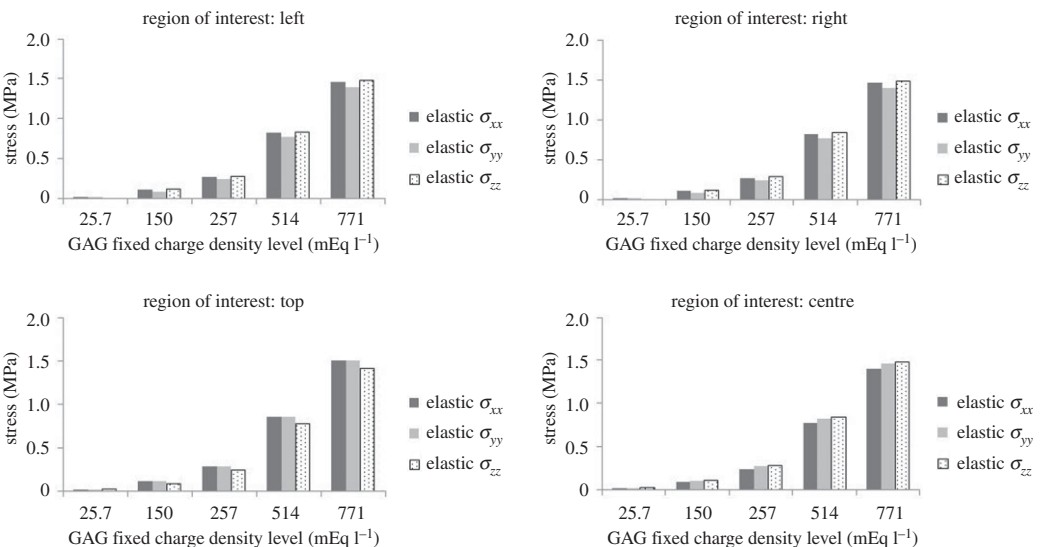

**Figure 8.** Maximum elastic stresses $\sigma_{xx}^e$, $\sigma_{yy}^e$ and $\sigma_{zz}^e$ in four ROIs for GAG inclusion FCD at 25.7, 150, 257, 514 and 771 mEq l$^{-1}$.

## 3.2. External bath osmolarity comparison

Similar to the FCD comparison for the GAG inclusion, five additional simulations were performed to study the influences of the external bath osmolarity on skin oedema. The skin FCD was set to 25.7 mEq l$^{-1}$, while the FCD for the GAG inclusion was set to 150 mEq l$^{-1}$. The external bath osmolarity was varied: 50, 100, 280, 500 and 1000 mOsm l$^{-1}$ levels. The total displacement and Von Mises stress comparison can be seen in figure 10.

The skin swelled more (increased displacement and increased stress) with increased sodium within the GAG inclusion, i.e. lower external bath osmolarity.

Additionally, four ROIs were defined and the maximum elastic tensile stresses $\sigma_{xx}^e$, $\sigma_{yy}^e$ and $\sigma_{zz}^e$ were computed. Figure 11 shows the maximum elastic stress at different GAG osmolarity levels for all four ROIs.

Figure 11 shows that the maximum tensile stress occurred at the top of the surrounding tissue where it interfaced with the GAG inclusion, and the maximum value was 0.31 MPa for both $\sigma_{xx}^e$ and $\sigma_{yy}^e$ when the external bath osmolarity was 50 mOsm l$^{-1}$. The elastic tensile stresses decreased with increased external bath osmolarity values.

Osmotic pressure within the GAG inclusion and the interstitial fluid pressure in the surrounding tissue are displayed in figure 12.

The osmotic pressure within the GAG inclusion decreased further during the swelling when the sodium content increased (lower external bath osmolarity). The interstitial fluid pressure in the surrounding tissue increased towards lower osmolarity as well.

# 4. Discussion and conclusion

## 4.1. Physiology and clinical application

In this study, the influences of negative charges (defined as FCD) associated with the GAG inclusion and the external bath osmolarity on the swelling of the skin tissue were investigated. A broader perspective of

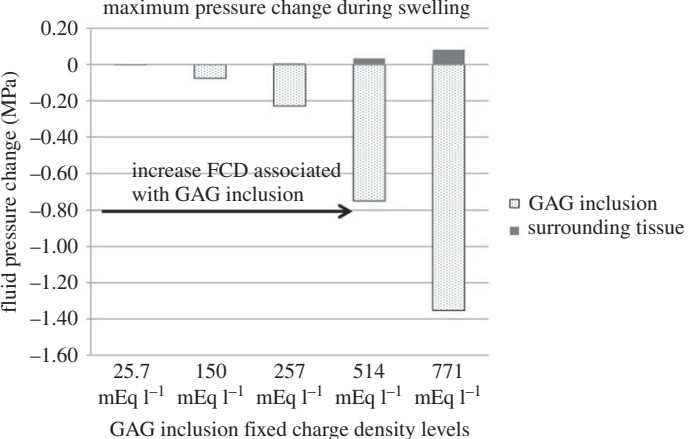

**Figure 9.** Maximum pressure change during swelling at FCD levels associated with the GAG inclusion of 25.7, 150, 257, 514 and 771 mEq l$^{-1}$.

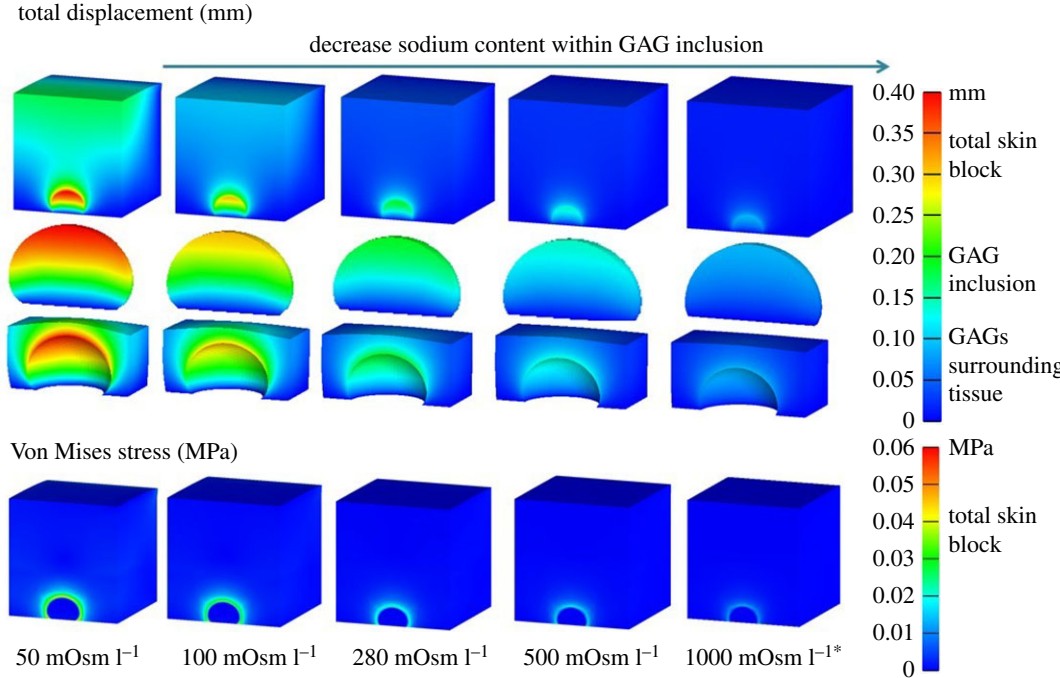

**Figure 10.** Comparison of the influences of osmolarity in external bath on the total displacement (mm) and Von Mises stress (MPa). From left to right, five simulations were taken when the external bath osmolarity was set to 50, 100, 280, 500 and 1000 mOsm l$^{-1}$. The largest deformation and highest stress were found when the osmolarity was 50 mOsm l$^{-1}$. The displacement and stress decreased with the increased GAG osmolarity. (*The external bath osmolarity is defined in FEBio; the lower value indicates higher sodium content within the inflamed tissue.)

this research was to relate blood pooling and inflammation to skin oedema and obtain the tissue deformation and stress states. Accomplishing this provided insights into the formation and development of venous ulcers.

While the osmolarity within the interstitial space of the tissue changed due to the accumulation of GAGs and sodium, the model defined the *external* bath osmolarity. Thus, the bath osmolarity was changed to induce an osmotic pressure difference between the external bath of the tissue and the tissue interstitium to simulate the water transfer across the membrane.

Although others have proposed models of wound development [14–16], our model was the first to investigate factors associated with *wound formation*. The tissue stress and pressure analyses conducted in this study provided an improved understanding of internal changes that are likely to occur during the formation of a venous ulcer.

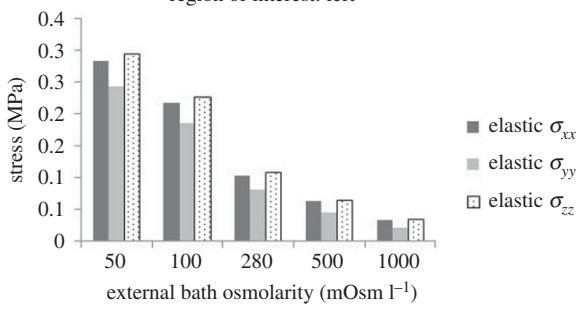

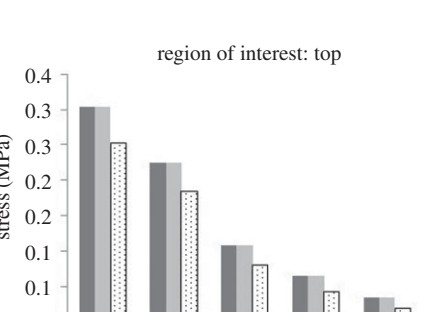

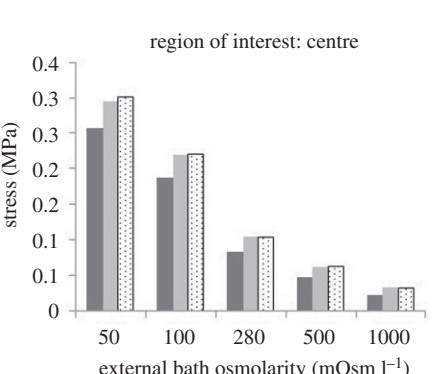

**Figure 11.** Maximum elastic stresses $\sigma_{xx}^{e}$, $\sigma_{yy}^{e}$ and $\sigma_{zz}^{e}$ in four ROIs for external bath osmolarity at 50, 100, 280, 500 and 1000 mOsm l$^{-1}$.

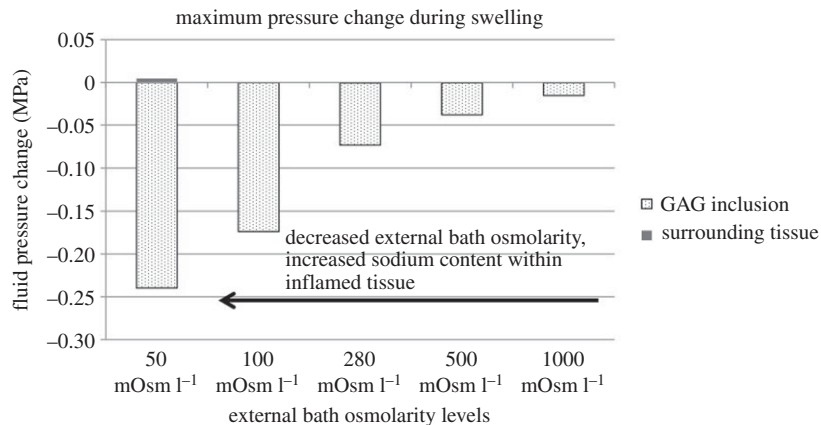

**Figure 12.** Maximum pressure change before and after swelling at different GAG inclusion external bath osmolarity levels.

This model suggested that regardless of the FCD or external osmolarity, the highest stresses were localized at the interface between the accumulated GAGs and the surrounding tissue. Our simulation results indicated that tissue damage was likely to first occur in the deep tissue and then propagate to the superficial level, resulting in open wounds. The model presented here may also be useful in the study of tissue stresses for other diseases that involve swelling, such as intestinal third spacing, lymph node oedema or brain oedema.

The elastic tensile stresses at the interface between the GAG inclusion and surrounding tissue reached the highest stress at 1.5 MPa when the GAG inclusion FCD was set to 771 mEq l$^{-1}$, while the skin FCD was at the normal level and both the GAG inclusion and skin tissue had normal osmolarity values. In the literature, a wide range of ultimate tensile strength (UTS) associated with skin tissue has been reported. Reports ranged between 0.1 and 40 MPa [66–68]. In consideration of the formation of venous ulcers, patients who develop these ulcers have compromised vascular systems and experience inflammation and oedema in their lower leg. The skin is weakened and unable to sustain pressure and stress [69,70]. Therefore, we chose to consider the UTS reported by Zhou *et al.* [71] (0.25–1.0 MPa), which

was at the lower end of the reported UTS range, because it is likely to be more representative of these patients. Based on these thresholds, our results suggest that the tissue is likely to tear with higher FCD and the lower external bath osmolarity values. The stress exceeded 0.25 MPa UTS reported in the literature for several of our simulations.

Simulation results also showed that when the GAG inclusion FCD was at 150 mEq l$^{-1}$, while holding other factors constant, the osmotic pressure change already exceeded the so-called safety factor for prevention of oedema, which was 15 mm Hg [50]. Therefore, we predict that oedema first ensues prior to the FCD level of 150 mEq l$^{-1}$. The increased 'drop' of osmotic pressure within the GAG inclusion at a higher FCD level and a lower external bath osmolarity value indicated more water would be transported from the external bath into the tissue leading to increased oedema and a higher risk for venous ulcers [52,72].

The results for the GAG inclusion FCD comparisons showed the interstitial fluid pressure in the surrounding tissue increased to 9.75 mm Hg (0.0013 MPa) when the GAG inclusion FCD was at 150 mEq l$^{-1}$ and increased to 47.25 mm Hg (0.0063 MPa) when the GAG inclusion FCD was at 257 mEq l$^{-1}$. The results from the external bath osmolarity comparisons showed that when the external bath osmolarity was reduced from 100 to 50 mOsm l$^{-1}$, the change of interstitial fluid pressure in the surrounding tissue during GAG swelling also increased, in this case from 3.35 mm Hg (0.0004 MPa) to 33.94 mm Hg (0.0045 MPa). This trend of increased stress and swelling with decreased external bath osmolarity has also been observed in experimental studies in other areas [73,74]. Studies reported that an increase in 10 mm Hg in the interstitial fluid pressure reduced capillary blood flow by half, causing tissue ischaemia and potentially leading to tissue necrosis [75–77] and ulcer formation [13,76].

By combining the GAG inclusion FCD and external bath osmolarity comparative results, a more detailed pathological explanation for venous ulcer formation was obtained. Physiologically, when skin becomes inflamed, the sodium content increases as well as the GAG content to regulate the inflammation [26–28]. The increased sodium content within the GAG inclusion (lower external bath osmolarity) and the increased GAG content (higher FCD) lead to increased skin swelling, as supported by the results of this study. When considering the GAG inclusion, the further decreased (more negative) osmotic pressure caused larger intracellular oedema [52], increasing the likelihood of venous ulceration [5,22]. As for the surrounding tissues, increased interstitial fluid pressure is known to result in increased skin ischaemia [75,78] and increased tissue elastic stress [44,79]; both ultimately lead to tissue necrosis and tissue damage and the formation of skin ulcers [80]. The simulation results presented in this study provide insights into the physiological pathway for the development of venous ulcers. These findings are reflected in an updated version of our initially hypothesized pathway (figure 13).

The results provide more detailed information to the hypothesis proposed by researchers with regard to the pathology of venous ulcers [1]. As illustrated in figure 13, the hypothesized pathophysiology for venous ulcer formation begins with blood pooling in the lower leg, causing inflammation, leading to skin oedema and resulting in skin ulceration. In the model, we observe that when GAG and sodium are increased, as reflected by the inflammatory process within the tissue, they cause an increase in deformation and higher fluid pressure, thus promoting more fluid transport across the membrane (i.e. increased permeability), resulting in skin oedema. Additionally, the high stress and pressure within the model show that oedema reaches levels sufficient to produce stresses capable of causing tissue damage and necrosis. Therefore, the model supports that blood pooling in the lower leg and sequential skin oedema and inflammation can be illustrated by the proliferation of sodium and GAG accumulation, and ulcer formation as well as tissue damage can be further explained by the microenvironment.

In summary, the work presented here simulates the pathological events that occur during inflammation and skin oedema, and indicates tissue damage that can lead to ulcer formation. This study is the first of its kind for venous ulcer formation, and indicates that further research into the role of GAGs and sodium in venous ulcer formation is warranted.

## 4.2. Limitations

The authors note limitations of this study. First, only the role of GAG and sodium was studied in the process of venous ulcer formation. Wound formation is a complex biomechanical and biochemical process; other contributors to ulcer formation, such as growth factor activation, and vessel compliance still need to be explored. In addition, it is true that increased extracellular pressure can itself induce intracellular signals that change the biology of the cells, adding another layer of complexity [81].

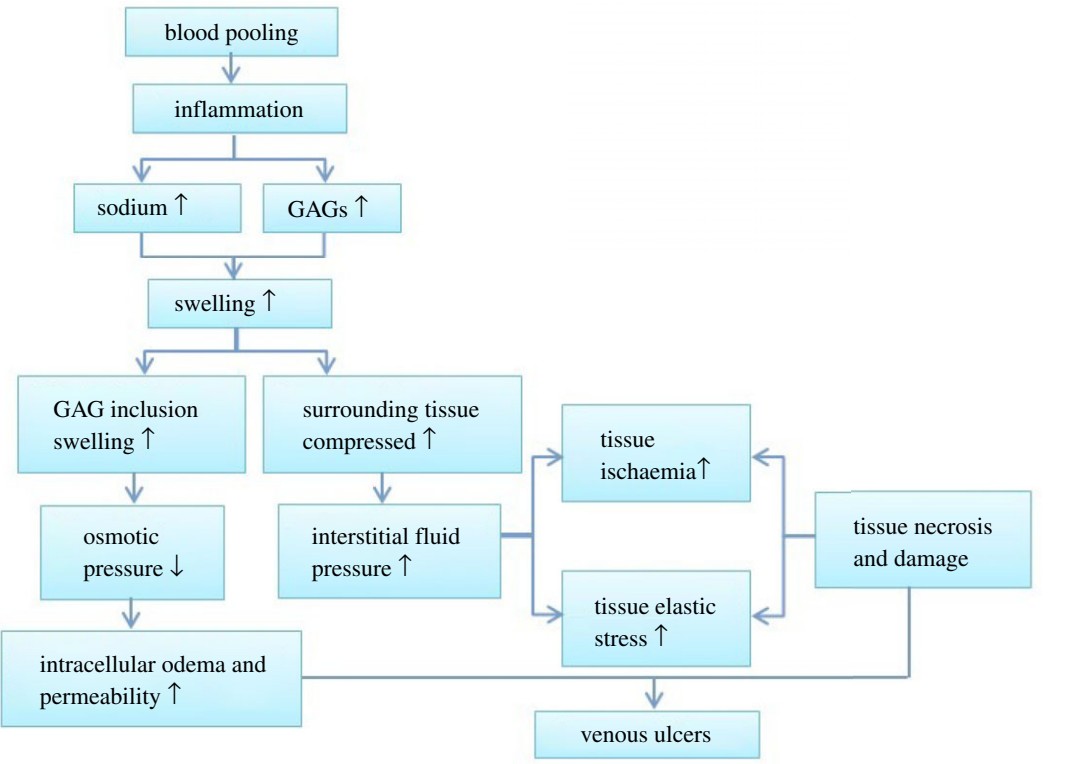

**Figure 13.** The detailed physiological pathways predicted by model simulations for venous ulcer formation. ↑ indicates increase, while ↓ refers to decrease.

Another limitation was the lack of literature describing the boundary values (upper and lower limits) of FCD values associated with the GAG inclusion as well as external bath osmolarity values for inflamed skin. In this study, a value of 30 times the normal skin GAG FCD level was used as the highest upper limit for the GAG inclusion FCD, and the external bath osmolarity was varied within the range reported by Azeloglu *et al.* [34]. Because of these uncertainties, five different FCD levels and five GAG external bath osmolarity levels were studied. Experimental studies are needed to document upper and lower boundary limits for these parameters. Nevertheless, this model laid a solid foundation for further parametric studies to explore and determine the critical FCD levels in the GAG inclusion and external bath osmolarity values for tissue damage.

The work presented here contributes significantly to our understanding of the role of FCDs associated with pooling GAG and changes in external bath osmolarity associated with oedema and swelling.

Data accessibility. Data are available from the Dryad Digital Repository: https://doi.org/10.5061/dryad.8tp4q6d [82].
Authors' contributions. W.P., S.R. and T.R.B. contributed to the design and execution of the study. W.P. performed the model simulations. M.D.B. contributed medical expertise on ulcer formation and clinical interpretations. S.R. and T.R.B. provided direction of the simulations, parameter selection and data analysis. All authors drafted, revised, critically reviewed and approved the final manuscript.
Competing interests. The authors have no competing interests.
Funding. We received no funding for this study.

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
