## [Reviewer comments · Royal Society Open Science]

Review History

RSOS-182076.R0 (Original submission)

Review form: Reviewer 1

Is the manuscript scientifically sound in its present form?

Yes

Are the interpretations and conclusions justified by the results?

Yes

Is the language acceptable?

Yes

Is it clear how to access all supporting data?

Yes

Do you have any ethical concerns with this paper?

No

Have you any concerns about statistical analyses in this paper?

No

Recommendation?

Accept as is

Comments to the Author(s)

I commend you on your work. I do not have any comments or questions.

Review form: Reviewer 2

Is the manuscript scientifically sound in its present form?

No

Are the interpretations and conclusions justified by the results?

No

Is the language acceptable?

No

Is it clear how to access all supporting data?

No

Do you have any ethical concerns with this paper?

No

Have you any concerns about statistical analyses in this paper?

I do not feel qualified to assess the statistics

Recommendation?

Reject

Comments to the Author(s)

Reviewer Comments for Manuscript RSOS-182076, "INFLUENCES OF SODIUM AND GLYCOSAMINOGLYCANS ON SKIN EDEMA AND THE POTENTIAL FOR ULCERATION: A FINITE ELEMENT APPROACH."

Overall:

1. The manuscript is poorly structured and reads like a BS or MS thesis, not a journal paper.
2. The manuscript is often in passive voice. Use active voice where possible.

Specifically (P = page, l = line):

P2-l41: Why is there a two-orders-of-magnitude range in the reported ultimate tensile stress of skin?

P6-Fig1: Why does it say "More Swelling"? There is no swelling earlier in the flow chart.

P6-l27: "a computational model for the skin is required" This is not true. This is the approach that

the authors selected, but it's not the only possibility.

P7-118: Why use a three-layer model? There are 2-5 layer models available in the literature. How does the choice of model affect the results and conclusions?

P7-125: How is the geometry of the inclusion determined and justified? Could this be determined by imaging approaches?

P7-137-40: "Boundary conditions were applied to all surfaces except the top surface. Zero displacement was used as the constraint for all surfaces (except the top)." Are these surfaces fixed in all dofs? If so, this is likely far, far too stiff. These should only be fixed normal to the plane.

P8-Fig3: This is mislabeled as Figure 1. Was a mesh sensitivity analysis performed?

P12-13-8: What is the justification for this approach? Why not use a design of experiments approach based on the physiological ranges, or something similar?

P12-116-21: How is this paper relevant to the current work?

P12-132-39: Are these values reasonable?

P13-Eq6: This is a biphasic implementation in FEBio. What are the models and elements used?

P14-16: Maximum tensile/compressive stress is usually referred to as the first/third principal stress.

P14-121: Why characterize the results using the Von Mises stress? What is the physiological meaning of this stress measure when comparing the results to the ultimate tensile stress?

P15-143: The results at the selected ROIs seems likely to derive from dependence of the mesh and may not be real.

P17-Fig7: The selection of σ_{xx} , σ_{yy} , and σ_{zz} is not meaningful, these measures are arbitrary as they depend on the choice of coordinate system.

P23-18: "which was at the lower end of the reported UTS range is likely to be more representative of these patients in comparison to the higher values reported" What is the justification for this statement?

P25-Fig12: Don't repeat the original figure. Isn't most of (b) inferable by intuition?

Review form: Reviewer 3

Is the manuscript scientifically sound in its present form?

Yes

Are the interpretations and conclusions justified by the results?

Yes

Is the language acceptable?

Yes

Is it clear how to access all supporting data?

Yes

Do you have any ethical concerns with this paper?

No

Have you any concerns about statistical analyses in this paper?

No

Recommendation?

Major revision is needed (please make suggestions in comments)

Comments to the Author(s)

This manuscript addresses the role of GAGs and sodium content on skin edema and increased tissue stresses as a potential mechanism for tissue damage and ulceration.

This is a well-designed basic science study and it is easy to read. The findings are interesting but perhaps not unexpected, considering the observations from previous studies on other biological tissues. I believe the paper still does add to the limited body of data regarding edema and potential mechanisms of skin ulceration. Below are some specific comments.

Page 2, lines 41-46: This is not a full sentence.

Page 4, line 55: The cited papers are a review and another finite element study. Is there any previous study with experimental evidence? A few suggestions are:

Nguyen et al. JOR 30(1):95-102, 2012

Bezci et al., J Biomech Eng 137(10):101007, 2015

Safa et al., Biomech 61(16):18-25

Page 5, line 18: Did these studies report how much sodium content increased in the inflamed tissue? If so, please include the reported values in parentheses. For example, a previous study reported a 3% increase in sodium content of the skin in women with lipedema (Crescenzi, 2018).

Page 5, line 30: "GAGS" should be "GAGs".

Page 7: Please specify the width of the layers and report the number of elements in the models. Did you use any constraints to connect the top layer to the underlying layer? Did you run a mesh sensitivity analysis to ensure that the mesh density was adequate for the simulations?

Page 14, lines 40-43: Please include a quantitative comparison of the changes in the maximum stress with an increase in FCD.

Pages 15-16: I suggest authors to move this paragraph to the methods section. This paragraph does not present any results.

Page 17: Please comment on the trends in stresses (i.e., nonlinear or linear) with an increase in GAGs FCD.

Page 21: Can you report the equation for the best-fit line to provide a functional relationship between fluid pressure change and external bath osmolarity?

Page 25: The updated schematic for the physiological pathway for venous ulcer formation shows an increase in permeability due to tissue swelling. Please elaborate on the schematic and support the claims with some experimental evidence from your work and previous studies.

Page 26: The physiological osmolarity for normal skin was reported to be ~280 mOsm/L. Would you expect to observe tissue shrinking or swelling if tissue sections were immersed in a hypotonic solution (i.e., 50 and 100 mOsm/L solutions)? Please elaborate on the initial configuration and its effect on the results presented in this study.

Review form: Reviewer 4

Is the manuscript scientifically sound in its present form?

No

Are the interpretations and conclusions justified by the results?

Yes

Is the language acceptable?

Yes

Is it clear how to access all supporting data?

Yes

Do you have any ethical concerns with this paper?

No

Have you any concerns about statistical analyses in this paper?

No

Recommendation?

Major revision is needed (please make suggestions in comments)

Comments to the Author(s)

Please see attached file "Comments on Pan et al.pdf" for details (Appendix A).

Decision letter (RSOS-182076.R0)

28-Mar-2019

Dear Dr Bush,

The editors assigned to your paper ("INFLUENCES OF SODIUM AND GLYCOSAMINOGLYCANS ON SKIN EDEMA AND THE POTENTIAL FOR ULCERATION: A FINITE ELEMENT APPROACH") have now received comments from reviewers. We would like you to revise your paper in accordance with the referee and Associate Editor suggestions which can be found below (not including confidential reports to the Editor). Please note this decision does not guarantee eventual acceptance.

Please submit a copy of your revised paper before 20-Apr-2019. Please note that the revision deadline will expire at 00.00am on this date. If we do not hear from you within this time then it will be assumed that the paper has been withdrawn. In exceptional circumstances, extensions may be possible if agreed with the Editorial Office in advance. We do not allow multiple rounds of revision so we urge you to make every effort to fully address all of the comments at this stage. If deemed necessary by the Editors, your manuscript will be sent back to one or more of the original reviewers for assessment. If the original reviewers are not available, we may invite new reviewers.

To revise your manuscript, log into <http://mc.manuscriptcentral.com/rsos> and enter your

Author Centre, where you will find your manuscript title listed under "Manuscripts with Decisions." Under "Actions," click on "Create a Revision." Your manuscript number has been appended to denote a revision. Revise your manuscript and upload a new version through your Author Centre.

- Data accessibility

If you wish to submit your supporting data or code to Dryad (<http://datadryad.org/>), or modify your current submission to dryad, please use the following link:
<http://datadryad.org/submit?journalID=RSOS&manu=RSOS-182076>

- Competing interests

- Authors' contributions

AB carried out the molecular lab work, participated in data analysis, carried out sequence alignments, participated in the design of the study and drafted the manuscript; CD carried out the statistical analyses; EF collected field data; GH conceived of the study, designed the study,

coordinated the study and helped draft the manuscript. All authors gave final approval for publication.

- Acknowledgements

- Funding statement

on behalf of Dr Derek Abbott (Associate Editor) and Professor R. Kerry Rowe (Subject Editor)
openscience@royalsociety.org

Comments to Author:

Reviewers' Comments to Author:

Reviewer: 1

Comments to the Author(s)

I commend you on your work. I do not have any comments or questions.

Reviewer: 2

Comments to the Author(s)

Reviewer Comments for Manuscript RSOS-182076, "INFLUENCES OF SODIUM AND GLYCOSAMINOGLYCANS ON SKIN EDEMA AND THE POTENTIAL FOR ULCERATION: A FINITE ELEMENT APPROACH."

Overall:

1. The manuscript is poorly structured and reads like a BS or MS thesis, not a journal paper.
2. The manuscript is often in passive voice. Use active voice where possible.

Specifically (P = page, l = line):

P2-141: Why is there a two-orders-of-magnitude range in the reported ultimate tensile stress of skin?

P6-Fig1: Why does it say "More Swelling"? There is no swelling earlier in the flow chart.

P6-127: "a computational model for the skin is required" This is not true. This is the approach that the authors selected, but it's not the only possibility.

P7-118: Why use a three-layer model? There are 2-5 layer models available in the literature. How does the choice of model affect the results and conclusions?

P7-125: How is the geometry of the inclusion determined and justified? Could this be determined by imaging approaches?

P7-137-40: "Boundary conditions were applied to all surfaces except the top surface. Zero displacement was used as the constraint for all surfaces (except the top)." Are these surfaces fixed in all dofs? If so, this is likely far, far too stiff. These should only be fixed normal to the plane.

P8-Fig3: This is mislabeled as Figure 1. Was a mesh sensitivity analysis performed?

P12-13-8: What is the justification for this approach? Why not use a design of experiments approach based on the physiological ranges, or something similar?

P12-116-21: How is this paper relevant to the current work?

P12-132-39: Are these values reasonable?

P13-Eq6: This is a biphasic implementation in FEBio. What are the models and elements used?

P14-16: Maximum tensile/compressive stress is usually referred to as the first/third principal stress.

P14-121: Why characterize the results using the Von Mises stress? What is the physiological meaning of this stress measure when comparing the results to the ultimate tensile stress?

P15-143: The results at the selected ROIs seems likely to derive from dependence of the mesh and may not be real.

P17-Fig7: The selection of σ_{xx} , σ_{yy} , and σ_{zz} is not meaningful, these measures are arbitrary as they depend on the choice of coordinate system.

P23-18: "which was at the lower end of the reported UTS range is likely to be more representative of these patients in comparison to the higher values reported" What is the justification for this statement?

P25-Fig12: Don't repeat the original figure. Isn't most of (b) inferable by intuition?

Reviewer: 3

Comments to the Author(s)

This manuscript addresses the role of GAGs and sodium content on skin edema and increased tissue stresses as a potential mechanism for tissue damage and ulceration.

This is a well-designed basic science study and it is easy to read. The findings are interesting but perhaps not unexpected, considering the observations from previous studies on other biological tissues. I believe the paper still does add to the limited body of data regarding edema and potential mechanisms of skin ulceration. Below are some specific comments.

Page 2, lines 41-46: This is not a full sentence.

Page 4, line 55: The cited papers are a review and another finite element study. Is there any previous study with experimental evidence? A few suggestions are:

Nguyen et al. JOR 30(1):95-102, 2012

Bezci et al., J Biomech Eng 137(10):101007, 2015

Safa et al., Biomech 61(16):18-25

Page 5, line 18: Did these studies report how much sodium content increased in the inflamed tissue? If so, please include the reported values in parentheses. For example, a previous study reported a 3% increase in sodium content of the skin in women with lipedema (Crescenzi, 2018).

Page 5, line 30: "GAGS" should be "GAGs".

Page 7: Please specify the width of the layers and report the number of elements in the models. Did you use any constraints to connect the top layer to the underlying layer? Did you run a mesh sensitivity analysis to ensure that the mesh density was adequate for the simulations?

Page 14, lines 40-43: Please include a quantitative comparison of the changes in the maximum stress with an increase in FCD.

Pages 15-16: I suggest authors to move this paragraph to the methods section. This paragraph does not present any results.

Page 17: Please comment on the trends in stresses (i.e., nonlinear or linear) with an increase in GAGs FCD.

Page 21: Can you report the equation for the best-fit line to provide a functional relationship between fluid pressure change and external bath osmolarity?

Page 25: The updated schematic for the physiological pathway for venous ulcer formation shows an increase in permeability due to tissue swelling. Please elaborate on the schematic and support the claims with some experimental evidence from your work and previous studies.

Page 26: The physiological osmolarity for normal skin was reported to be ~280 mOsm/L. Would you expect to observe tissue shrinking or swelling if tissue sections were immersed in a hypotonic solution (i.e., 50 and 100 mOsm/L solutions)? Please elaborate on the initial configuration and its effect on the results presented in this study.

Reviewer: 4

Comments to the Author(s)

Please see attached file "Comments on Pan et al.pdf" for details.

Author's Response to Decision Letter for (RSOS-182076.R0)

See Appendix B.

RSOS-182076.R1 (Revision)

Review form: Reviewer 1

Is the manuscript scientifically sound in its present form?

Yes

Are the interpretations and conclusions justified by the results?

Yes

Is the language acceptable?

Yes

Is it clear how to access all supporting data?

Not Applicable

Do you have any ethical concerns with this paper?

No

Have you any concerns about statistical analyses in this paper?

No

Recommendation?

Accept as is

Comments to the Author(s)

It appears that you have addressed reviewers comments/questions. A final manuscript review is necessary to make sure word tense is consistent, ie, at times I notice a flip from past to present which is incorrect; also, verify that ALL references to figures match with text, eg, page 16, I think you should be referring to figure 6, not 5.

Review form: Reviewer 3

Is the manuscript scientifically sound in its present form?

Yes

Are the interpretations and conclusions justified by the results?

Yes

Is the language acceptable?

Yes

Is it clear how to access all supporting data?

Yes

Do you have any ethical concerns with this paper?

No

Have you any concerns about statistical analyses in this paper?

No

Recommendation?

Accept with minor revision (please list in comments)

Comments to the Author(s)

The authors have satisfactorily responded to all my questions and made the necessary changes to the manuscript. I advise the authors to proofread the manuscript before publication. This paper still has some grammar issues, which need to be addressed.

For example, on page 28 - line 23, it should be "only the role of GAGs and sodium was studied...".

Review form: Reviewer 4

Is the manuscript scientifically sound in its present form?

Yes

Are the interpretations and conclusions justified by the results?

Yes

Is the language acceptable?

Yes

Is it clear how to access all supporting data?

Yes

Do you have any ethical concerns with this paper?

No

Have you any concerns about statistical analyses in this paper?

No

Recommendation?

Accept with minor revision (please list in comments)

Comments to the Author(s)

Thank you for the care you took in addressing my concerns.

Decision letter (RSOS-182076.R1)

23-May-2019

Dear Dr Bush:

On behalf of the Editors, I am pleased to inform you that your Manuscript RSOS-182076.R1 entitled "INFLUENCES OF SODIUM AND GLYCOSAMINOGLYCANS ON SKIN EDEMA AND THE POTENTIAL FOR ULCERATION: A FINITE ELEMENT APPROACH" has been accepted for publication in Royal Society Open Science subject to minor revision in accordance with the referee suggestions. Please find the referees' comments at the end of this email.

The reviewers and Subject Editor have recommended publication, but also suggest some minor revisions to your manuscript. Therefore, I invite you to respond to the comments and revise your manuscript.

- Ethics statement

- Data accessibility

<http://datadryad.org/submit?journalID=RSOS&manu=RSOS-182076.R1>

- Competing interests

- Authors' contributions

- Acknowledgements

- Funding statement

Because the schedule for publication is very tight, it is a condition of publication that you submit the revised version of your manuscript before 01-Jun-2019. Please note that the revision deadline will expire at 00.00am on this date. If you do not think you will be able to meet this date please let me know immediately.

on behalf of Dr Derek Abbott (Associate Editor) and R. Kerry Rowe (Subject Editor)
openscience@royalsociety.org

Reviewer comments to Author:

Reviewer: 4

Comments to the Author(s)

Thank you for the care you took in addressing my concerns.

Reviewer: 3

Comments to the Author(s)

The authors have satisfactorily responded to all my questions and made the necessary changes to the manuscript. I advise the authors to proofread the manuscript before publication. This paper still has some grammar issues, which need to be addressed.

For example, on page 28 - line 23, it should be "only the role of GAGs and sodium was studied...".

Reviewer: 1

Comments to the Author(s)

It appears that you have addressed reviewers comments/questions. A final manuscript review is necessary to make sure word tense is consistent, ie, at times I notice a flip from past to present which is incorrect; also, verify that ALL references to figures match with text, eg, page 16, I think you should be referring to figure 6, not 5.

Author's Response to Decision Letter for (RSOS-182076.R1)

See Appendix C.

Decision letter (RSOS-182076.R2)

03-Jun-2019

Dear Dr Bush,

I am pleased to inform you that your manuscript entitled "INFLUENCES OF SODIUM AND GLYCOSAMINOGLYCANS ON SKIN EDEMA AND THE POTENTIAL FOR ULCERATION: A FINITE ELEMENT APPROACH" is now accepted for publication in Royal Society Open Science.

on behalf of Dr Derek Abbott (Associate Editor) and R. Kerry Rowe (Subject Editor)
openscience@royalsociety.org

Associate Editor Comments to Author (Dr Derek Abbott):

Reviewer comments to Author:

Follow Royal Society Publishing on Twitter: [@RSocPublishing](https://twitter.com/RSocPublishing)
Follow Royal Society Publishing on Facebook:
<https://www.facebook.com/RoyalSocietyPublishing.FanPage/>
Read Royal Society Publishing's blog: <https://blogs.royalsociety.org/publishing/>

Appendix A

Comments on Pan et al., Influences of sodium and glycosaminoglycans on skin edema and the potential for ulceration: a finite element approach

The hypothesis and associated physiological mechanisms are well-described. Separating out the physiological impact of GAGs and osmolarity is interesting. However, the problem statement is unclear and requires clarification and reorganization in section 2.

The major flaw is that the theoretical formulation is not included. The manuscript should explicitly lay out the (1) governing equations and (2) mathematical description of the boundary conditions. It's very difficult to understand the foundation of the numerical simulation without it, particularly since some of the terminology is confusing (discussed below). All of the theory should be laid out before diving into material parameters and design of the parametric study. The discussion in section 2.4 provides a useful differentiation between the physiological effects caused by Na and GAGs; however it would be better placed within the theoretical formulation, either at the end of section 2.1 or as a stand-alone section before the current section 2.2.

The computational domain is well-described by Fig 3 (mistakenly labeled Fig 1 – also the color scheme in Fig 1(c) and (d) should match the other images of the domain (Fig 1(a) and (b) and Fig 4)). However, matching them to the physiological analogs which are being modeled is not so obvious from Figs 3 and 4 and the text. This is exacerbated by the fact that some of the terminology is confusing. The term “GAGs inclusion” is not terribly helpful since GAGs are molecules and have a characteristic distribution throughout the domain. Perhaps using “GAGs-induced inclusion” or simply “inclusion” would be an option that better describes the spatial nature of what you are describing.

The authors distinguish between the “external bath” and “interstitial fluid”, but it's unclear where these fluid domains exist. Fig 4 attempts to define the external bath in relation to the computational domain, but, at least for me, it muddies the waters further. The external bath looks like a 2-dimensional plane behind the 3D skin block as though it exists at the back face of the block in contact with all 3 layers of skin and extends beyond the domain of the epidermis. I don't think that this is the intent. A better visual and written description that describes where these fluid domains reside in relation to the skin block is needed, as well as how the fluid in the skin block, the external bath, and the interstitial space can interact. The text describes how water from the external bath can be drawn “across the membrane” into the interstitial space around p5, line 35, but it doesn't describe what the membrane is. Also, the spatial link between the skin block and the veins is needed since the veins are the ultimate source of the excess fluid in the skin block. A description of the theoretical formulation might have helped to infer the relationships, but I think that you are going to lose your readers unless this is explicitly laid out.

In Fig 5, do these images represent a steady-state solution or are they snapshots at a particular time?

The discussion of results on p 23 is quite good and provides some quantitative estimates of threshold values of their parameters that are associated with pathological changes.

A discussion of grid resolution should be included to confirm that the numerical domain is sufficiently resolved and is essential for establishing credibility. This can be either a brief discussion in the manuscript or a more rigorous description in supplemental material. Grid resolution is particularly important when a new model does not include any validation against experiment and/or other numerical simulations. This may be because there is currently nothing in the literature against which to

validate. In the limitations section, the authors describe the need for experimental data to bound a couple of their parameters. Are there other data that would help to validate your model?

The authors chose to use a volume fraction within the range reported in the literature. Do the authors think that the solution is sensitive to volume fraction?

Minor issues:

P2 li 41. Sentence beginning “Thus suggesting” makes this a sentence fragment. Suggest using “This suggests”

P4 last sentence before section 1.1: the verbs have mixed tense.

In section 1.1, 1st paragraph, the description of wound angiogenesis appears the second time that the term is used. It would be better to include “(formation of new blood vessels)” the first time that the term is used.

In Figs 8 and 11, it’s hard to see the edges of the boxes with dotted fill. Suggest that you use a solid color or include a bounding box around the dotted region. In the legend, should GAGs be “GAGs inclusion” for consistency? Also, there is a typo in “Surrounding tissue”.

In Fig 9 title, there is an extra “the” before “defined”.

P11 li 46, typo: largestt magnitude of swelling

Appendix B

Reviewer: 1-----

Comments to the Author(s)

I commend you on your work. I do not have any comments or questions.

Thank you for taking the time to read our manuscript and the positive feedback.

Reviewer: 2 -----

Reviewer Comments for Manuscript RSOS-182076, "INFLUENCES OF SODIUM AND GLYCOSAMINOGLYCANS ON SKIN EDEMA AND THE POTENTIAL FOR ULCERATION: A FINITE ELEMENT APPROACH."

Overall:

1. The manuscript is poorly structured and reads like a BS or MS thesis, not a journal paper.
2. The manuscript is often in passive voice. Use active voice where possible.

Thank you, we reviewed and edited the manuscript for structure and voice. Specifically, we modified the abstract, introduction and discussion as well as implemented specific structural revisions requested by the other reviewers. We recognize that the background section is not typical for all types of publications, but because of the limited research conducted on venous ulcers, we thought it was important for the readers to be familiar with the general physiology associated with venous ulcer formation prior to discussing the model and simulation results.

Specifically (P = page, l = line):

P2-l41: Why is there a two-orders-of-magnitude range in the reported ultimate tensile stress of skin?

The reviewer is correct. The literature of skin testing indicates a large range in UTS. The authors did not conduct these studies that are referenced therefore we cannot comment on their methods and accuracy. We can, however speak to skin as a biological tissue. It is a well-documented fact that the elasticity of skin changes over time; thus aged skin will be stiffer (lower strain with higher rupture stress) due to this reduction in elasticity than young skin. Additionally, different body regions yield slightly different mechanical behavior (cheek vs. heel) and the skin sample preparation methods also play a role in the outcomes of the testing. Specifically, the 0.1-40 MPa value was reported from literature where it showed the ultimate tensile stress of a sample from an 80 year old individual which was almost 100 times higher than that of a sample from 9 month old child. These ultimate tensile stress values can be found in the references listed here:

[1] Gallagher AJ, Ni Anniadh A, Kruyere K, Ottenio M, Xie H, Gilchrist MD. Dynamic tensile properties of human skin. 2012 IRCOBI Conf 2012:494–502. doi:Irc-12-59.

[2] Flynn C, Taberner A, Nielsen P. Modeling the mechanical response of in vivo human skin under a rich set of deformations. *Ann Biomed Eng* 2011;39:1935–46. doi:10.1007/s10439-011-0292-7.

[3] Ní Annaidh A, Bruyère K, Destrade M, Gilchrist MD, Otténio M. Characterization of the anisotropic mechanical properties of excised human skin. *J Mech Behav Biomed Mater* 2012;5:139–48. doi:10.1016/j.jmbbm.2011.08.016.

P6-Fig1: Why does it say “More Swelling”? There is no swelling earlier in the flow chart.

Thank you, we can see how this may be confusing. We have removed the word “more” and just left swelling in the box. (Now Figure 2)

P6-l27: “a computational model for the skin is required” This is not true. This is the approach that the authors selected, but it’s not the only possibility.

Thank you for this comment, this sentence has been rewritten and is also copied here:

“In order to validate this pathway and characterize the detailed mechanical changes in the skin tissue, a computational model for the skin was developed.”

P7-l18: Why use a three-layer model? There are 2-5 layer models available in the literature. How does the choice of model affect the results and conclusions?

The three layers in the model represent the three layers of the skin: epidermis, dermis, and hypodermis. Models with other layers include more layers in the Epidermis (Stratum Corneum) or referring the base layer beneath dermis as subcutaneous fat. The models that use the stratum corneum as an individual layer usually combine the rest of the epidermis with the dermis layer as the epidermis layer itself is extremely thin. The subcutaneous fat is a synonym for hypodermis. Therefore, we believe the three-layer model best represents skin anatomy and material properties. Additionally, all three layers of the skin were bonded together without frictional force between the layers, thus the skin block behaved as a whole and the differences between each layer consisted of the thickness and the material properties (Young’s modulus and Poisson’s ratio).

P7-l25: How is the geometry of the inclusion determined and justified? Could this be determined by imaging approaches?

We do not have any experimental information that would lead us in one direction or another for the geometry of the GAGs inclusion. However, we selected the sphere which would generate the most conservative values in stress. A square would have corner stress

concentrations and an ellipse would have higher curvatures yielding higher stresses. However, the volume of the inclusion was an estimation based on the reported GAGs content in skin. GAGs consisted about 0.1%~0.3% of dry weight in skin and 0.37%~0.42% wet weight in skin [1,2]. The skin model in the study had the volume of 125 mm^3 , and the GAGs inclusion had a volume of 0.23 mm^3 . This provided us a volume fraction of GAGs inclusion at 0.18% at the baseline condition. Changes in geometry for the GAGs concentration will be included in future work.

- [1] Salbach, Juliane, Tilman D Rachner, Sandra Franz, Jan-christoph Simon, and Lorenz C Hofbauer. 2012. "Regenerative Potential of Glycosaminoglycans for Skin and Bone." *Journal of Molecular Medicine* 90 (6): 625–35. doi:10.1007/s00109-011-0843-2.
- [2] Wiig, H., and M. A. Swartz. 2012. "Interstitial Fluid and Lymph Formation and Transport: Physiological Regulation and Roles in Inflammation and Cancer." *Physiological Reviews* 92 (3): 1005–60. doi:10.1152/physrev.00037.2011.

P7-I37-40: "Boundary conditions were applied to all surfaces except the top surface. Zero displacement was used as the constraint for all surfaces (except the top)." Are these surfaces fixed in all dofs? If so, this is likely far, far too stiff. These should only be fixed normal to the plane.

We did not fix the surfaces in all dofs; we apologize for the confusion. All surfaces were constrained somewhat, except for the top surface that represented the portion of the epidermis in contact with air. We identified constraints that took into account the symmetry of the model about the XZ plane, which was constrained only normally to the surface (see Figure below). We also considered that we are analyzing a small portion of skin, extrapolated from a large semi-infinite in vivo geometry (when compared to the dimensions of our model). This led us to constrain the displacements in both the normal and shear direction in the back, front, and right faces, which are ideally in contact with the rest of the skin. Finally, since we were mostly concerned with investigating the effects of upward swelling (namely toward the surface of the skin), as opposed to downward (which in this case would mean internal swelling) we decided to constrain the bottom surface in every direction.

We have clarified the manuscript as follows (page 9):

Boundary conditions were applied to represent the fact that the model is symmetric with respect to the YZ-plane, see Figure 7, and is isolated from a larger skin plane. Namely, we constrained the normal displacement on the plane of symmetry, and normal and shear displacements on the planes in contact with surrounding skin. Because we are interested in upward / outward swelling, the top surface, representing the side of the epidermis in contact with air, was allowed free swelling, while the bottom surface, representing the side of the hypodermis on contact with surrounding tissue, was constrained in every direction.

P8-Fig3: This is mislabeled as Figure 1. Was a mesh sensitivity analysis performed?

We apologize for the mislabeling of Figure 3, that has been corrected. The authors did perform a mesh sensitivity analysis. The element numbers was increased from 28896 to 37296 (a 29% increase) to generate a more refined mesh, and the results (displacement and stress) did not show any difference within the 4 decimals output by FEBio. This information has been added to the manuscript (page 9).

P12-I3-8: What is the justification for this approach? Why not use a design of experiments approach based on the physiological ranges, or something similar?

Unfortunately, there are no reported lower or upper limit values for the FCD or osmolarity in the inflamed skin for the patient population likely to develop venous ulcers (i.e., individuals with vascular disease). This work is specifically focused on this population and was used to obtain insights of what physiological events happen from the initial blood pooling to skin edema and finally ulceration. This work was innovative in its aim to quantify how GAGs increased presence and possible pooling could lead to skin tissue breakdown associated with blood pooling and swelling as observed in the venous ulcer population.

P12-I16-21: How is this paper relevant to the current work?

Given that there is a lack of reported values on the range of osmolarity for skin inflammation and skin ulceration, we broadened our literature review to consider all types of soft biological tissues. Azeloglu et al., conducted the research with aorta ring (a soft biological tissue) and the osmolarity value range from this paper provides a viable reference for understanding the

osmolarity limits associated with biological tissues. It would be ideal to have a reference specific to skin and our patient population, but that does not yet exist.

P12-I32-39: Are these values reasonable?

The FCD values for normal skin are reported in the literature (Wiig et al., 2000; Wiig et al., 2012) we used this as the starting point and then expanded the range. Additionally, one author used an FCD of 150 in her previously published work (Roccabianca et al., 2014; Roccabianca et al., 2014). This is the first work to study the relationship between the GAGs inclusion and skin ulceration. Therefore, we used the normal range and expanded from there since no information regarding the range of the FCD for inflamed skin is present in the literature.

The references are listed below:

*Wiig H, Swartz MA. Interstitial Fluid and Lymph Formation and Transport: Physiological Regulation and Roles in Inflammation and Cancer. *Physiol Rev* 2012;92:1005–60. doi:10.1152/physrev.00037.2011.*

*Wiig H, Reed RK, Tenstad O. Interstitial fluid pressure, composition of interstitium, and interstitial exclusion of albumin in hypothyroid rats. *Am J Physiol Heart Circ Physiol* 2000;278:H1627–39.*

*Roccabianca S, Bellini C, Humphrey JD. Computational modelling suggests good, bad and ugly roles of glycosaminoglycans in arterial wall mechanics and mechanobiology. *J R Soc Interface* 2014;11:20140397. doi:10.1098/rsif.2014.0397.*

*Roccabianca S, Ateshian GA, Humphrey JD. Biomechanical roles of medial pooling of glycosaminoglycans in thoracic aortic dissection. *Biomech Model Mechanobiol* 2014;13:13–25. doi:10.1007/s10237-013-0482-3.*

P13-Eq6: This is a biphasic implementation in FEBio. What are the models and elements used?

We developed the mesh using the Hypermesh software, and we used hexahedral elements HEX8 (added to page 9). We employed the biphasic theory at equilibrium, which is already implemented in FEBio, namely the Donna Equilibrium Swelling material. This material represents a porous material, with a charged solid matrix, and an external solution environment that contains monovalent ions. To describe the solid matrix, we used a neo-Hookean material model for all layers of skin, but we specified different mechanical parameters for each layer (as described on page 12 in section 2.3.1) This is a must for this material description, since the Donnan equilibrium theory is based on the existence of a solid matrix that “resists” the swelling. We are aware that an isotropic description of skin is an approximation, however not having access to in house directional mechanical information we felt that this approximation was reasonable for the time being. We will in the future

consider the possibility of using a more precise description for the solid matrix (for example considering a fiber reinforced material description).

Finally, the material we consider is porous, however this does not represent a biphasic analysis per se, these results are only valid once the fluid equilibrium is reached, namely we are not analyzing what happens in the transient response. We believe this is a reasonable approximation, since we are considering this to be a pathological condition that leads to PUs, as opposed to a transient condition (for example as associated with inflammation due to trauma).

P14-l6: Maximum tensile/compressive stress is usually referred to as the first/third principal stress.

The authors agree with this statement. However, the authors chose the use of tensile/compressive stress as they are more descriptive and these terms are used by clinicians and surgeons while the first/third principal stress is not commonly used outside of engineering. So, to support readability of researchers from an array of backgrounds, we elected to use these terms.

We have added a clarification statement in the manuscript as follows (page 12):

Given that the shear stress component is zero in Eq. 11, the maximum tensile/compressive stress could also be referred to as the 1st and 3rd principal stresses, respectively. The authors chose to use the term tensile stress throughout the manuscript as it is a more descriptive term.

P14-l21: Why characterize the results using the Von Mises stress? What is the physiological meaning of this stress measure when comparing the results to the ultimate tensile stress?

The Von Mises stress here was reported to visually understand the critical areas where high stress localization was observed. To relate these data to the physiology, further detailed stress analyses were conducted with the principal stresses in the regions that high Von Mises was displayed. We later compared these principal stresses to the ultimate tensile stress (Figures 8 and 11).

P15-l43: The results at the selected ROIs seems likely to derive from dependence of the mesh and may not be real.

The results at the selected ROIs were the averaged value of the elements in the selected regions. The ROI represents the area where localized high stress occurred. As noted in an earlier response, we did conduct a sensitivity analysis. The elements number was increased from 28896 to 37296 (a 29% increase) to generate a more refined mesh, and the results (displacement and stress) did not show any difference within the 4 decimals output by FEBio.

P17-Fig7: The selection of σ_{xx} , σ_{yy} , and σ_{zz} is not meaningful, these measures are arbitrary as they depend on the choice of coordinate system.

To keep the description of results consistent for all ROIs, we chose to use the global reference system and refer to σ_{xx} , σ_{yy} , and σ_{zz} during analysis. The coordinate system was displayed in Figure 6. At different regions of interest, the stress direction is different. Such as σ_{yy} is perpendicular at the right and left sides and parallel to the top.

P23-l8: "which was at the lower end of the reported UTS range is likely to be more representative of these patients in comparison to the higher values reported" What is the justification for this statement?

The lower UTS ranges were derived from samples that came from 5–9 month porcine skin (Gallagher et al., 2012); porcine has been shown to be the closest to human skin material properties and have long been used as an alternative for excised human skin tissue (Groves et al., 2013; Ankersen et al., 1999). In consideration of the formation of venous ulcers, patients who develop these ulcers have compromised vascular systems and are experiencing inflammation and edema in their lower leg, causing decreased skin extensibility (Pierard et al., 2014). With edema, the skin is weakened and unable to sustain pressure and stress (Bansal et al., 2005)

The references for this statement are:

[1] Gallagher AJ, Ni Anniadh A, Kruyere K, Ottenio M, Xie H, Gilchrist MD. Dynamic tensile properties of human skin. 2012 IRCOB Conf 2012:494–502. doi:Irc-12-59.

[2] Groves, Rachel B., et al. "An anisotropic, hyperelastic model for skin: experimental measurements, finite element modelling and identification of parameters for human and murine skin." Journal of the mechanical behavior of biomedical materials 2013; 18: 167-180.

[3] Ankersen J, Birkbeck AE, Thomson RD, Vanezis P. Puncture resistance and tensile strength of skin simulants. Proceedings of the Institution of Mechanical Engineers, Part H: Journal of Engineering in Medicine. 1999;213(6):493-501.

[4] Bansal C, Scott R, Stewart D, Cockerell CJ. Decubitus ulcers: a review of the literature. Int J Dermatol 2005;44:805–10. doi:10.1111/j.1365-4632.2005.02636.x.

[5] Pierard GE, Paquet P, Piérard - Franchimont C. Skin viscoelasticity in incipient gravitational syndrome. Journal of cosmetic dermatology. 2014;13(1):52-5.

P25-Fig12: Don't repeat the original figure. Isn't most of (b) inferable by intuition?

Thank you for the comment the original figure has been removed.

Reviewer: 3 -----

This manuscript addresses the role of GAGs and sodium content on skin edema and increased tissue stresses as a potential mechanism for tissue damage and ulceration.

This is a well-designed basic science study and it is easy to read. The findings are interesting but perhaps not unexpected, considering the observations from previous studies on other biological tissues. I believe the paper still does add to the limited body of data regarding edema and potential mechanisms of skin ulceration. Below are some specific comments.

Thank you for the comment. Indeed, we agree and believe this work provides insights to the physiological events that are not possible to observe.

Page 2, lines 41-46: This is not a full sentence.

The sentence has been rewritten and also copied here:

The results suggested that both the edema and increased fluid pressure could reach a point of tissue damage and eventual ulcer formation.

Page 4, line 55: The cited papers are a review and another finite element study. Is there any previous study with experimental evidence? A few suggestions are:

Nguyen et al. JOR 30(1):95-102, 2012

Bezci et al., J Biomech Eng 137(10):101007, 2015 Safa et al., Biomech 61(16):18-25

During our literature review, we were unable to locate studies specifically associated with venous ulcers or skin tissue. Thank you for the suggested papers. We cited Bezci's work in the introduction (page 4 ref 31). We also reviewed all three manuscripts from Bezci et al., Nguyen et al., and Safa et al., and both papers provided a good justification for our model results. We have included Bezci's work together with the finite element study, and also added the other two references in the discussion section as follows:

The results from the external bath osmolarity comparisons show that when the external bath osmolarity was reduced from 100 to 50 mOsm/L, the change of interstitial fluid pressure in the surrounding tissue during GAGs swelling also increased, in this case from 3.35 mmHg (0.0004 MPa) to 33.94 mmHg (0.0045 MPa). This trend of increased stress and swelling with decreased external bath osmolarity has also been observed in experimental studies (Nguyen and Levenston 2012; Safa et al. 2017). Studies have reported that an increase of 10mmHg in the interstitial fluid pressure reduces capillary blood flow in half, causing tissue

ischemia, and potentially leading to tissue necrosis (Odland et al. 2004; Hargens et al. 1989; Wiese 1993) and ulcer formation (Hargens et al. 1989; McGee et al. 2009).

Page 5, line 18: Did these studies report how much sodium content increased in the inflamed tissue? If so, please include the reported values in parentheses. For example, a previous study reported a 3% increase in sodium content of the skin in women with lipedema (Crescenzi, 2018).

The two reference cited here discussed the relationship between inflammation and increased sodium, as well as the change in GAGs. Schwartz et al., reported increased osmolarity values associated with inflammation and Titze et al., reported skin Na⁺ increased from 140 mmol/L to 180~190 mmol/L with increased GAGs content. However, both of the references were experimental studies with controlled laboratory settings, instead of in vivo measurements.

We have added a sentence in the manuscript that specifically states the above (page 5)

In soft tissue inflammation, an increase in sodium, concurrent with GAG accumulation, has been reported to occur (Schwartz et al. 2009; Titze et al. 2004; Crescenzi et al. 2018). Titze et al., reported sodium increased from 140 mmol/L to 180~190 mmol/L with increased GAGs content (Titze et al. 2004), and Crescenzi et al., observed 3% increase in sodium content associated with lipedema (Crescenzi et al. 2018).

Thank you for suggesting this new reference, we have also included in our manuscript.

Page 5, line 30: "GAGS" should be "GAGs".

Thank you, this has been corrected.

Page 7: Please specify the width of the layers and report the number of elements in the models. Did you use any constraints to connect the top layer to the underlying layer? Did you run a mesh sensitivity analysis to ensure that the mesh density was adequate for the simulations?

Thank you for these questions. We have added this information to the manuscript (page 9). The width and depth of each layer is 5mm x 5mm. The total number of the elements is 28896. In these simulations, the layers were considered to be perfectly bonded with one another. We did perform a mesh sensitivity study where the elements number was increased from 28896 to 37296 (a 29% increase) to generate a more refined mesh, and the results (displacement and stress) did not show any difference within the 4 decimals output by FEBio

Page 14, lines 40-43: Please include a quantitative comparison of the changes in the maximum stress with an increase in FCD.

A detailed stress analysis with quantitative comparisons between changes in FCD and changes in maximum principal stress are available in Figure 8. The Von Mises stress was used in Figure 6 to provide a visual perspective of where the localized high stresses occurred, here we only showed the trend of increased stress with increased FCD.

The following was added to the text (page 19):

The highest stress was 400 times more when the FCD was 30 times higher than the normal value. When the GAGs inclusion FCD was at 150 mEq/L, the same value reported being used by Roccabianca et al., [37], the maximum elastic stress was 30 times higher than that at normal FCD value.

Pages 15-16: I suggest authors to move this paragraph to the methods section. This paragraph does not present any results.

Thank you for the suggestion. The reason why this paragraph was initially positioned here was because without the preliminary observation of the high stress localization, it was not possible to predetermine the Regions of Interest (ROIs). We have edited the manuscript and these statements have been moved to the methods section.

Page 17: Please comment on the trends in stresses (i.e., nonlinear or linear) with an increase in GAGs FCD.

We appreciate this question. Using the left ROI as an example, we plotted the stresses against GAGs FCD, it can be seen that the stresses had a nonlinear increase with GAGs FCD. We have included a statement of this trend in our discussion (page 26). This statement is also below:

“The maximum stress exhibited a nonlinear increase with the increase of the GAGs inclusion FCD.”

Stress with Increased FCD (Left ROI)

Page 21: Can you report the equation for the best-fit line to provide a functional relationship between fluid pressure change and external bath osmolarity?

Thank you for the comment. In order to find the best-fit trend line, the logarithm transform was performed and the fluid pressure change can be expressed with a polynomial fit of the log osmolarity values. The equations and R^2 values are reported in the figures below:

Surrounding Tissue Pressure Change

Page 25: The updated schematic for the physiological pathway for venous ulcer formation shows an increase in permeability due to tissue swelling. Please elaborate on the schematic and support the claims with some experimental evidence from your work and previous studies.

Thank you for this comment. Permeability by definition is the ability of the membrane wall to allow fluids to pass. When tissue is swollen, the pressure difference between the two sides of the membrane is increased, pushing the fluid transport through the membrane. Therefore, we have indicated permeability occurs in association with increased edema. We have included text in the manuscript noting this relationship (page 27). Additionally, several researchers have also discussed the relationship between edema and increased permeability [1-3]:

[1] Guo, Xiaomei, Yoram Lanir, and Ghassan S Kassab. 2007. "Effect of Osmolarity on the Zero-Stress State and Mechanical Properties of Aorta." *American Journal of Physiology. Heart and Circulatory Physiology* 293 (4): H2328–34. doi:10.1152/ajpheart.00402.2007.

[2] Langemo, D K. 1999. "Venous Ulcers: Etiology and Care of Patients Treated with Human Skin Equivalent Grafts." *Journal of Vascular Nursing* 17 (1): 6–11. <http://www.ncbi.nlm.nih.gov/pubmed/10362981>.

[3] McGee, Maria P, Michael Morykwas, Nicole Levi-Polyachenko, and Louis Argenta. 2009. "Swelling and Pressure-Volume Relationships in the Dermis Measured by Osmotic-Stress Technique." *American Journal of Physiology. Regulatory, Integrative and Comparative Physiology* 296 (6): R1907–13. doi:10.1152/ajpregu.90777.2008.

We have added the following to the manuscript (page 27):

We observed that when GAGs and sodium were increased, as reflected by the inflammatory process within the tissue, they caused an increase in deformation and higher fluid pressure, thus promoting more fluid transport across the membrane (i.e. increased permeability), resulting in skin edema.

Page 26: The physiological osmolarity for normal skin was reported to be ~280 mOsm/L. Would you expect to observe tissue shrinking or swelling if tissue sections were immersed in a hypotonic solution (i.e., 50 and 100 mOsm/L solutions)? Please elaborate on the initial configuration and its effect on the results presented in this study.

When the tissue is immersed in a hypotonic solution, which means the external bath has a lower sodium (Na⁺) concentration compared to inside the tissue, more water will be drawn from the external bath into the tissue due to the pressure/concentration difference. Therefore, when the tissue is in the hypotonic solution, it will swell. Here is a schematic presentation of this swelling process. Since two reviewers asked questions about this point, we have added this figure to the manuscript (Figure 1).

Reviewer: 4 -----

Comments on Pan et al., Influences of sodium and glycosaminoglycans on skin edema and the potential for ulceration: a finite element approach

The hypothesis and associated physiological mechanisms are well-described. Separating out the physiological impact of GAGs and osmolarity is interesting.

However, the problem statement is unclear and requires clarification and reorganization in section 2. The major flaw is that **the theoretical formulation is not included. The manuscript should explicitly lay out the (1) governing equations and (2) mathematical description of the boundary conditions.** It's very difficult to understand the foundation of the numerical simulation without it, particularly since some of the terminology is confusing (discussed below). All of the theory should be laid out before diving into material parameters and design of the parametric study.

The discussion in section 2.4 provides a useful differentiation between the physiological effects caused by Na and GAGs; however it would be better **placed within the theoretical formulation, either at the end of section 2.1 or as a stand-alone section before the current section 2.2.**

Thank you for the suggestions. Based on Reviewer 4's comments, we have reorganized Section 2 significantly. As requested, the theoretical formulation has been moved earlier in the manuscript.

The material model used here was a mixture of Donnan swelling and Neo-Hookean solids. Therefore, the governing equations for such biphasic material based on conservation of momentum and conservation of mass are

$$\text{div}\boldsymbol{\sigma} = -\nabla p + \text{div}\boldsymbol{\sigma}^s$$

$$\text{div}\left(\frac{\partial \mathbf{u}^s}{\partial t} + \mathbf{Q}\right) = 0$$

Where \mathbf{u}^s is the porous solid matrix displacement and \mathbf{Q} is the fluid flux relative to the solid matrix.

The boundary conditions are:

$$\mathbf{u}^s(0) = 0;$$

$$\mathbf{u}^s(t) = 0 \text{ for quarter symmetric boundaries}$$

$$p(0) = RT \sqrt{(C_0^F)^2 + (\bar{C}^*_0)^2} - \bar{C}^*_0 ;$$

$$Q(t) = 0$$

The fluid flux at the final stage of the equilibrium will be zero as the fluid transfer reach the balance. The symmetric boundaries have the displacement constraints as follows:

The computational domain is well-described by **Fig 3 (mistakenly labeled Fig 1 – also the color scheme in Fig 1(c) and (d) should match the other images of the domain (Fig 1(a) and (b) and Fig 4))**. However, matching them to the physiological analogs which are being modeled is not so obvious from Figs 3 and 4 and the text.

Thank you, this has been corrected. Figure 3 (a)~(d) and Figure 4 (now Figures 4 & 5) were obtained from the same model. Figure 4 (a) was displayed as a transparent image for the left half and the wireframe image displayed for the right half. We presented in this format because we thought it was easier to see the symmetry and the three layers could be easily seen and identified.

We see how the blue colors can be confusing, so we have modified the color scheme to be gray (epidermis), teal (dermis), and blue (hypodermis) so that this confusion is avoided.

The physiological analog can be seen in Figure 1, and we have labeled Figures 3, 4 and 5 with matching labels identifying the three layers of the skin, and the GAGs inclusion.

This is exacerbated by the fact that some of the terminology is confusing. The term “GAGs inclusion” is not terribly helpful since GAGs are molecules and have a characteristic distribution throughout the domain. **Perhaps using “GAGs-induced inclusion” or simply “inclusion” would be an option that better describes the spatial nature of what you are describing.**

The GAGs inclusion represents the volume of the tissue cells with accumulated GAG molecules. Although the GAGs were distributed throughout the entire skin tissue blocks, the tissue that contains the GAGs inclusion had a much higher GAGs concentration. This is the rationale as to why the specific region was called a “GAGs inclusion”. Such terminology was used in Roccabianca et al., (2014) and continued here. We have added a line in the text that clarifies and defines what is meant by the GAGs inclusion. We believe this will reduce the potential confusion; thank you for pointing this out (page 8).

The authors distinguish between the “external bath” and “interstitial fluid”, but it’s unclear where these fluid domains exist. Fig 4 attempts to define the external bath in relation to the computational domain, but, at least for me, it muddies the waters further. The external bath looks like a 2-dimensional plane behind the 3D skin block as though it exists at the back face of the block in contact with all 3 layers of skin and extends beyond the domain of the epidermis. I don’t think that this is the intent. **A better visual and written description that describes where these fluid domains reside in relation to the skin block is needed, as well as how the fluid in the skin block, the external bath, and the interstitial space can interact.** The text describes how **water from the external bath can be drawn “across the membrane” into the interstitial space around p5, line 35, but it doesn’t describe what the membrane is.**

Thank you for these comments. We have added a figure and modified a figure to address these concerns. We acknowledge that representing the 3D situation in 2D is not ideal and have modified the image so that the bath is clearly identified as being along all sides of the block. This figure was also modified to match the new color scheme of Figure 4.

A modified visual presentation of the external bath is:

In addition, another reviewer asked for the analog presentation of the water and GAGs movement during inflammation and swelling, so we have added the figure below as well.

Also, **the spatial link between the skin block and the veins is needed** since the veins are the ultimate source of the excess fluid in the skin block. A description of the theoretical formulation might have helped to infer the relationships, but I think that you are going to lose your readers unless this is explicitly laid out.

We have updated the Figure 3 and added the location of the vein. We also included a statement in the manuscript. (Page 7)

A schematic of the broader perspective of modeling blood pooling and skin edema through changes in GAGs and sodium content is located in Figure 2. As veins are located in the hypodermis layer of the skin, we expected the GAGs and sodium accumulation would first occur in this region.

In Fig 5, do these images represent a steady-state solution or are they snapshots at a particular time?

These images represent a steady state solution as provided in the boundary conditions of this model. The model reached an equilibrium at the end of each simulation. This is also explained in the “pressure drop” at the end of the simulation as the model is at a steady state, the pressure gradient required across the membrane is lowered.

The discussion of results on p 23 is quite good and provides some quantitative estimates of threshold values of their parameters that are associated with pathological changes.

A discussion of grid resolution should be included to confirm that the numerical domain is sufficiently resolved and is essential for establishing credibility. This can be either a brief discussion in the manuscript or a more rigorous description in supplemental material. Grid resolution is particularly important when a new model does not include any validation against experiment and/or other numerical simulations. **This may be because there is currently nothing in the literature against which to validate.** In the limitations section, the authors describe the need for experimental data to bound a couple of their parameters. **Are there other data that would help to validate your model?**

We have performed a mesh sensitivity study where the element numbers was increased from 28896 to 37296 (a 29% increase) to generate a more refined mesh, and the results (displacement and stress) did not show any difference within the 4 decimals output by FEBio. This proved the sufficiency of grid resolution in this study. This information has been added to the manuscript (page 9).

The study presented here was one of the first to apply the Donnan swelling model to the study of venous ulcers. The adopted methodology and model set up is comparable to previous studies reported in the literature for other soft tissue studies (Roccabianca, Bellini, and Humphrey 2014; Ateshian et al. 2011; Yu, Malakpoor, and Huyghe 2018). Although these studies were on other tissues, the swelling observed in our work followed similar trends as those reported in these publications.

The authors chose to use a **volume fraction** within the range reported in the literature. Do the authors think that the **solution is sensitive to volume fraction?**

The volume fraction we used was for the model fluid volume fraction. This was a physiological value that has been reported between 0.70-0.85 for a hydrated tissue. The fluid volume fraction, together with the solid volume fraction describes the porosity of the skin. For this parameter we do have physiological values included within a small range. Based on the equations:

$$p = RT\sqrt{(C^F)^2 + (\bar{C}^*)^2} - \bar{C}^*$$

$$C^F = \frac{\varphi_0^w C_0^F}{J - 1 + \varphi_0^w}$$

The difference between Max. and Min. volume fraction φ_0^w is only 0.15, where $R^*T=2.577E-03$, which is a very small scale. Therefore, the values selected in our volume fraction range will not significantly impact the calculated pressure change.

Minor issues:

P2 li 41. Sentence beginning “Thus suggesting” makes this a sentence fragment. Suggest using “This suggests”

Thank you for the comment, this has been rewritten as follows:

The results suggested that both the edema and increased fluid pressure could reach a point of tissue damage and eventual ulcer formation.

P4 last sentence before section 1.1: the verbs have mixed tense.

This has been corrected:

The overall goal of this study was to develop a model that simulated the inflammation, **determine** the internal stresses and pressure of the skin tissue, and **provide** an improved understanding of these mechanisms and their association with venous ulcer formation.

In section 1.1, 1st paragraph, the description of wound angiogenesis appears the second time that the term is used. It would be better to include “(formation of new blood vessels)” the first time that the term is used.

Thank you for the comment, the “formation of new blood vessels” has been moved up to when the wound angiogenesis first appeared.

In Figs 8 and 11, it’s hard to see the edges of the boxes with dotted fill. Suggest that you use a solid color or include a bounding box around the dotted region. In the legend, should GAGs be “GAGs inclusion” for consistency? Also, there is a typo in “Surrounduding tissue”.

Thank you for the comment.

Thank you, both Figures have been updated. A heavier line has been used as a bounding box for the dotted region and the typo has been fixed.

In Fig 9 title, there is an extra “the” before

“defined”.

P11 li 46, typo: largest magnitude of swelling

Thank you, both of these have been addressed

Appendix C

MICHIGAN STATE UNIVERSITY

May 31, 2019

Dear Editor –

We are pleased to submit our revised manuscript entitled “Influences of Sodium and Glycosaminoglycans (GAGs) on Skin Edema and the Potential for Ulceration: A Finite Element Approach” for publication in the *Royal Society Open Science*.

As requested, we have carefully gone through the manuscript and made appropriate corrections for tense and grammar. In addition, we sought out a communications editor with a degree in English and he too reviewed the manuscript.

We also confirmed that all necessary statements are included at the end of the manuscript (Ethics, Data Accessibility, Author Contributions, Competing Interests, Funding, and Acknowledgements).

Sincerely,

COLLEGE OF
ENGINEERING

Department of
Mechanical
Engineering

Tamara Reid Bush, Ph.D.
Associate Professor, ASME Fellow
Department of Mechanical Engineering
2555 Engineering Building
Michigan State University
East Lansing, MI 48824
517-353-9544, reidtama@msu.edu

Michigan State University
2555 Engineering
Building
East Lansing, Michigan
48824-1226